# Train Hard, Fight Easy:
# Robust Meta Reinforcement Learning

**Ido Greenberg**
Technion, Nvidia Research
gido@campus.technion.ac.il

**Shie Mannor**
Technion, Nvidia Research
shie@ee.technion.ac.il

**Gal Chechik**
Bar Ilan University, Nvidia Research
gchechik@nvidia.com

**Eli Meirom**
Nvidia Research
emeirom@nvidia.com

## Abstract

A major challenge of reinforcement learning (RL) in real-world applications is the variation between environments, tasks or clients. Meta-RL (MRL) addresses this issue by learning a meta-policy that adapts to new tasks. Standard MRL methods optimize the average return over tasks, but often suffer from poor results in tasks of high risk or difficulty. This limits system reliability since test tasks are not known in advance. In this work, we define a robust MRL objective with a controlled robustness level. Optimization of analogous robust objectives in RL is known to lead to both **biased gradients** and **data inefficiency**. We prove that the gradient bias disappears in our proposed MRL framework. The data inefficiency is addressed via the novel Robust Meta RL algorithm (***RoML***). RoML is a meta-algorithm that generates a robust version of any given MRL algorithm, by identifying and over-sampling harder tasks throughout training. We demonstrate that RoML achieves robust returns on multiple navigation and continuous control benchmarks.

## 1 Introduction

Reinforcement learning (RL) has achieved impressive results in a variety of applications in recent years, including cooling systems control [Luo et al., 2022] and conversational chatbots [Cohen et al., 2022]. A significant challenge in extending this success to mass production is the variation between instances of the problem, e.g., different cooling systems or different chatbot end-users. Meta-RL (MRL) addresses this challenge by learning a "meta-policy" that quickly adapts to new tasks [Thrun and Pratt, 1998, Finn et al., 2017]. In the examples above, MRL would maximize the average return of the adapted policy for a new cooling system or a new end-user.

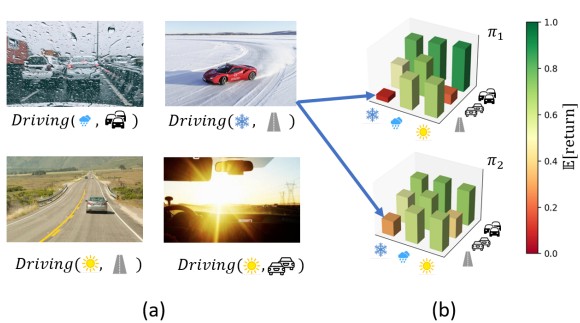

(a)                    (b)

Figure 1: (a) An illustration of driving tasks, characterized by various weather conditions and traffic density. (b) The returns of two meta-policies $\pi_1, \pi_2$ on these tasks. $\pi_1$ has a higher average return, but $\pi_2$ is more robust to high-risk tasks. The task space is discretized only for illustration purposes.

However, optimizing the return of the average client might not suffice, as certain clients may still experience low or even negative returns. If 10% of the clients report poor performance, it may deter potential clients from adopting the new technology – even if its average return is high. This highlights

37th Conference on Neural Information Processing Systems (NeurIPS 2023).

the need for MRL systems that provide robust returns across tasks. Robustness is further motivated by risk sensitivity in many natural RL applications, such as medical treatment [Yu et al., 2021] and driving [Shalev-Shwartz et al., 2016]. For example, as illustrated in Fig. 1, an agent should drive safely at *any* road profile – even if at some roads the driving would be more cautious than necessary.

A common approach for risk-averse optimization is the max-min objective [Collins et al., 2020]; in MRL, that would mean searching for a meta-policy with the highest expected-return in the worst possible task. This expresses the most extreme risk aversion, which only attends to the one worst case out of all the possible outcomes. Furthermore, in certain problems, worst cases are inevitable (e.g., in certain medical treatments, a fatal outcome cannot be avoided), thus optimizing the minimum return might not provide any meaningful result. A more general objective is the average return over the worst $\alpha$ quantiles ($0 \leq \alpha \leq 1$), also known as the Conditional Value-at-Risk (CVaR). Notice that the CVaR is a generalization of both the mean (for $\alpha = 1$) and the minimum (for $\alpha = 0$).

CVaR is a coherent risk measure used for risk management in various fields [Filippi et al., 2020], including banking regulation [Acerbi and Szekely, 2014] and RL [Tamar et al., 2015, Hiraoka et al., 2019]. In this work, we extend CVaR optimization to MRL, by replacing the standard MRL objective

$$\underset{\pi}{\mathrm{argmax}}\, J^{\theta}(R), \quad J^{\theta}(R) = \mathbb{E}_{\tau,R}[R], \tag{1}$$

with a CVaR objective, which measures the robustness of a policy to high-risk tasks:

$$\underset{\pi}{\mathrm{argmax}}\, J_{\alpha}^{\theta}(R), \quad J_{\alpha}^{\theta}(R) = \mathtt{CVaR}_{\tau}^{\alpha}\left[\mathbb{E}_{R}[R]\right]. \tag{2}$$

In both equations, $\tau$ is a random task and $R$ is the random return of policy $\pi_{\theta}$ in $\tau$. Intuitively, the CVaR return expresses robustness to the selected task, in analogy to robustness to the realized model in standard RL. To further motivate Eq. (2), note that CVaR optimization in RL is equivalent to robust optimization under uncertain perturbations [Chow et al., 2015].

In Section 4, we follow the standard approach of policy gradient (PG) for $\mathtt{CVaR}_{\alpha}$ optimization in RL, and apply it to MRL. That is, for every batch of $N$ trajectories, we apply the learning step to the $\alpha N$ trajectories with the lowest returns. This standard approach, CVaR-PG, is known to suffer from a major limitation: in an actor-critic framework, the critic leads to a biased gradient estimator – to the extent that it may point to the opposite direction [Tamar et al., 2015]. This limitation is quite severe: many CVaR-PG implementations [Tamar et al., 2015, Greenberg et al., 2022] rely on vanilla PG without a critic (REINFORCE, Williams [1992]); others pay the price of gradient bias – in favor of advanced actor-critic methods that reduce the gradient variance [Rajeswaran et al., 2017].

This limitation is particularly concerning in *meta* RL, where high complexity and noise require more sophisticated algorithms than REINFORCE. Fortunately, **Section 4 eliminates this concern: in MRL, in contrast to RL, the CVaR policy gradient is proven to remain unbiased regardless of the choice of critic.** Hence, our proposed method – CVaR Meta Learning (*CVaR-ML*) – can be safely applied on top of any MRL algorithm. This makes CVaR-ML a *meta-algorithm*: given an arbitrary MRL algorithm, CVaR-ML generates a robust version of it.

Nevertheless, in CVaR optimization methods, another source of gradients variance and sample inefficiency is the large proportion of data not being utilized. Every iteration, we rollout trajectories for $N$ tasks, but only use $\alpha N$ of them for training. To mitigate this effect, we introduce in Section 5 the Robust Meta RL algorithm (*RoML*). RoML assumes that tasks can be selected during training. It learns to identify tasks with lower returns and over-samples them. By training on high-risk tasks, the meta-agent learns policies that are robust to them without discarding data. Hence, **RoML increases the sample efficiency by a factor of up to $\alpha^{-1}$**. Unlike common adversarial methods, which search for the worst-case sample (task) that minimizes the return [Collins et al., 2020], RoML lets the user specify the desired level of robustness $\alpha$, and addresses the entire $\alpha$-tail of the return distribution.

We test our algorithms on several domains. Section 6.1 considers a navigation problem, where both CVaR-ML and RoML obtain better CVaR returns than their risk-neutral baseline. Furthermore, they learn substantially different navigation policies. Section 6.2 considers several continuous control environments with varying tasks. These environments are challenging for CVaR-ML, which entirely fails to learn. Yet, RoML preserves its effectiveness and consistently improves the robustness of the returns. In addition, Section 6.3 demonstrates that under certain conditions, RoML can be applied to supervised settings as well – providing robust supervised meta-learning.

As a meta-algorithm, in each experiment RoML improves the robustness of its baseline algorithm – using the same hyper-parameters as the baseline. The *average* return is also improved in certain experiments, indicating that even the risk-neutral objective of Eq. (1) may benefit from robustness.

**Contribution:** (a) We propose a principled CVaR optimization framework for robust meta-RL. While the analogous problem in standard RL suffers from biased gradients and data inefficiency, we (b) prove theoretically that MRL is immune to the former, and (c) address the latter via the novel Robust Meta RL algorithm (RoML). Finally, (d) we demonstrate the robustness of RoML experimentally.

## 2   Related Work

**Meta-RL** for the **average task** is widely researched, including methods based on gradients [Finn et al., 2017, Gupta et al., 2018], latent memory [Zintgraf et al., 2019, Rakelly et al., 2019] and offline meta learning [Dorfman et al., 2020, Pong et al., 2022]. It is used for applications ranging from robotics [Nagabandi et al., 2018] to education [Wu et al., 2021]. Adversarial meta learning was studied for minimax optimization of the **lowest-return task**, in supervised meta learning [Collins et al., 2020, Goldblum et al., 2020] and MRL [Lin et al., 2020]. Other works studied the robustness of MRL to distributional shifts [Mendonca et al., 2020, Ajay et al., 2022]. However, the **CVaR task** objective has not been addressed yet in the framework of MRL.

**Risk-averse RL.** In *standard* RL, risk awareness is widely studied for both safety [García and Fernández, 2015, Greenberg and Mannor, 2021] and robustness [Derman et al., 2020]. CVaR specifically was studied using PG [Tamar et al., 2015, Rajeswaran et al., 2017, Hiraoka et al., 2019, Huang et al., 2021a], value iteration [Chow et al., 2015] and distributional RL [Dabney et al., 2018, Schubert et al., 2021, Lim and Malik, 2022]. CVaR optimization was also shown equivalent to mean optimization under robustness [Chow et al., 2015], motivating robust-RL methods [Pinto et al., 2017, Godbout et al., 2021]. In this work, we propose a *meta-learning* framework and algorithms for CVaR optimization, and point to both similarities and differences from the standard RL setting.

**Sampling.** In Section 5, we use the cross-entropy method [de Boer et al., 2005] to sample high-risk tasks for training. The cross-entropy method has been studied in standard RL for both optimization [Mannor et al., 2003, Huang et al., 2021b] and sampling [Greenberg et al., 2022]. Sampling in RL was also studied for regret minimization in the framework of Unsupervised Environment Design [Dennis et al., 2020, Jiang et al., 2021]; and for accelerated curriculum learning in the framework of Contextual RL [Klink et al., 2020, Eimer et al., 2021]. By contrast, we address MRL (where the current task is unknown to the agent, unlike Contextual RL), and optimize the CVaR risk measure instead of the mean.

## 3   Preliminaries

**MRL.** Consider a set of Markov Decision Processes (MDPs) $\{(S, A, \tau, \mathcal{P}_\tau, \mathcal{P}_{0,\tau}, \gamma)\}_{\tau \in \Omega}$, where the distribution of transitions and rewards $\mathcal{P}_\tau$ and the initial state distribution $\mathcal{P}_{0,\tau}$ both depend on task $\tau \in \Omega$. The task itself is drawn from a distribution $\tau \sim D$ over a general space $\Omega$, and is not known to the agent. The agent can form a belief regarding the current $\tau$ based on the task history $h$, which consists of repeating triplets of states, actions and rewards [Zintgraf et al., 2019]. Thus, the meta-policy $\pi_\theta(a; s, h)$ $(\theta \in \Theta)$ maps the current state $s \in S$ and the history $h \in \prod(S \times A \times \mathbb{R})$ (consisting of state-action-reward triplets) to a probability distribution over actions.

A meta-rollout is defined as a sequence of $K \geq 1$ episodes of length $T \in \mathbb{N}$ over a single task $\tau$: $\Lambda = \{\{(s_{k,t}, a_{k,t}, r_{k,t})\}_{t=1}^T\}_{k=1}^K$. For example, in a driving problem, $\tau$ might be a geographic area or type of roads, and $\Lambda$ a sequence of drives on these roads. The return of the agent over a meta-rollout is defined as $R(\Lambda) = \frac{1}{K}\sum_{k=1}^K \sum_{t=0}^T \gamma^t r_{k,t}$, where $r_{k,t}$ is the (random variable) reward at step $t$ in episode $k$. Given a task $\tau$ and a meta-policy $\pi_\theta$, we denote by $P_\tau^\theta(x)$ the conditional PDF of the return $R$. With a slight abuse of notation, we shall use $P_\tau^\theta(\Lambda)$ to also denote the PDF of the meta-rollout itself. The standard MRL objective is to maximize the expected return $J^\theta(R) = \mathbb{E}_{\tau, R}[R]$.

While the meta policy $\pi_\theta(s, a; h)$ is history-dependent, it can still be learned using standard policy gradient (PG) approaches, by considering $h$ as part of an extended state space $\tilde{s} = (s, h) \in \tilde{S}$. Then,

the policy gradient can be derived directly:

$$\nabla_\theta J^\theta(R) = \int_\Omega D(z) \int_{-\infty}^{\infty} (x-b)\nabla_\theta P_z^\theta(x) \cdot dx \cdot dz, \tag{3}$$

where $D$ is a probability measure over the task space $\Omega$, $P_z^\theta(x)$ is the PDF of $R$ (conditioned on $\pi_\theta$ and $\tau = z$), and $b$ is any arbitrary baseline that is independent of $\theta$ [Agrawal, 2019]. While a direct gradient estimation via Monte Carlo sampling is often noisy, its variance can be reduced by an educated choice of baseline $b$. In the common actor-critic framework [Mnih et al., 2016], a learned value function $b = V(s; h)$ is used. This approach is used in many SOTA algorithms in deep RL, e.g., PPO [Schulman et al., 2017]; and by proxy, in MRL algorithms that rely on them, e.g., VariBAD [Zintgraf et al., 2019].

A major challenge in MRL is the extended state space $\tilde{S}$, which now includes the whole task history. Common algorithms handle the task history via a low-dimensional embedding that captures transitions and reward function [Zintgraf et al., 2019]; or using additional optimization steps w.r.t. task history [Finn et al., 2017]. Our work does not compete with such methods, but rather builds upon them: our methods operate as meta-algorithms that run on top of existing MRL baselines.

**CVaR-PG.** Before moving on to CVaR optimization in MRL, we first recap the common PG approach for standard (non-meta) RL. For a random variable $X$ and $\alpha$-quantile $q_\alpha(X)$, the CVaR is defined as $\texttt{CVaR}_\alpha(X) = \mathbb{E}\left[X \mid X \leq q_\alpha(X)\right]$. For an MDP $(S, A, P, P_0, \gamma)$, the CVaR-return objective is $\tilde{J}_\alpha^\theta(R) = \texttt{CVaR}_{R \sim P^\theta}^\alpha[R] = \int_{-\infty}^{q_\alpha^\theta(R)} x \cdot P^\theta(x) \cdot dx$, whose corresponding policy gradient is [Tamar et al., 2015]:

$$\nabla_\theta \tilde{J}_\alpha^\theta(R) = \int_{-\infty}^{q_\alpha^\theta(R)} (x - q_\alpha^\theta(R)) \cdot \nabla_\theta P^\theta(x) \cdot dx. \tag{4}$$

Given a sample of $N$ trajectories $\{\{(s_{i,t}, a_{i,t})\}_{t=1}^T\}_{i=1}^N$ with returns $\{R_i\}_{i=1}^N$, the policy gradient can be estimated by [Tamar et al., 2015, Rajeswaran et al., 2017]:

$$\nabla_\theta \tilde{J}_\alpha^\theta(R) \approx \frac{1}{\alpha N} \sum_{i=1}^N \mathbf{1}_{R_i \leq \hat{q}_\alpha^\theta} \cdot (R_i - \hat{q}_\alpha^\theta) \cdot \sum_{t=1}^T \nabla_\theta \log \pi_\theta(a_{i,t}; s_{i,t}), \tag{5}$$

where $\hat{q}_\alpha^\theta$ is an estimator of the current return quantile.

Notice that in contrast to mean-PG, in CVaR optimization the baseline *cannot* follow an arbitrary critic, but should approximate the total return quantile $q_\alpha^\theta(R)$. Tamar et al. [2015] showed that any baseline $b \neq q_\alpha^\theta(R)$ inserts bias to the CVaR gradient estimator, potentially to the level of pointing to the opposite direction (as discussed in Appendix A.1 and Fig. 6). As a result, CVaR-PG methods in RL either are limited to basic REINFORCE with a constant baseline [Greenberg et al., 2022], or use a critic for variance reduction at the cost of biased gradients [Rajeswaran et al., 2017].

Another major source of gradient-variance in CVaR-PG is its reduced sample efficiency: notice that Eq. (5) only exploits $\approx \alpha N$ trajectories out of each batch of $N$ trajectories (due to the term $\mathbf{1}_{R_i \leq \hat{q}_\alpha^\theta}$), hence results in estimation variance larger by a factor of $\alpha^{-1}$.

## 4  CVaR Optimization in Meta-Learning

In this section, we show that **unlike standard RL, CVaR-PG in MRL permits a flexible baseline *without* presenting biased gradients**. Hence, policy gradients for *CVaR* objective in *MRL* is substantially different from both *mean*-PG in MRL (Eq. (3)) and CVaR-PG in *RL* (Eq. (4)).

To derive the policy gradient, we first define the policy value per task and the tail of tasks.

**Definition 1.** The value of policy $\pi_\theta$ in task $\tau$ is denoted by $V_\tau^\theta = \mathbb{E}_{R \sim P_\tau^\theta}[R]$. Notice that $V_\tau^\theta$ depends on the random variable $\tau$. We define the $\alpha$-tail of tasks w.r.t. $\pi_\theta$ as the tasks with the lowest values: $\Omega_\alpha^\theta = \{z \in \Omega \mid V_z^\theta \leq q_\alpha(V_\tau^\theta)\}$.

**Assumption 1.** To simplify integral calculations, we assume that for any $z \in \Omega$ and $\theta \in \Theta$, $R$ is a continuous random variable (i.e., its conditional PDF $P_z^\theta(x)$ has no atoms). We also assume that $v(z) = V_z^\theta$ is a continuous function for any $\theta \in \Theta$.

**Theorem 1** (Meta Policy Gradient for CVaR). Under Assumption 1, the policy gradient of the CVaR objective in Eq. (2) is

$$\nabla_\theta J_\alpha^\theta(R) = \int_{\Omega_\alpha^\theta} D(z) \int_{-\infty}^{\infty} (x - b) \nabla_\theta P_z^\theta(x) \cdot dx \cdot dz, \tag{6}$$

where $b$ is *any* arbitrary baseline independent of $\theta$.

*Proof intuition (the formal proof is in Appendix A).* In RL, the CVaR objective measures the $\alpha$ lowest-return trajectories. When the policy is updated, the cumulative probability of these trajectories changes and no longer equals $\alpha$. Thus, the new CVaR calculation must add or remove trajectories (as visualized in Fig. 6 in the appendix). This adds a term in the gradient calculation, which causes the bias in CVaR-PG. By contrast, in MRL, the CVaR measures the $\alpha$ lowest-return *tasks* $\Omega_\alpha^\theta$. Since the task distribution does not depend on the policy, the probability of these tasks is not changed – but only the way they are handled by the agent (Fig. 7). Thus, no bias term appears in the calculation. Note that $\Omega_\alpha^\theta$ does change throughout the meta-learning – due to changes in task *values* (rather than task probabilities); this is a different effect and is not associated with gradient bias. □

According to Theorem 1, the CVaR PG in MRL permits *any* baseline $b$. As discussed in Section 3, this flexibility is necessary, for example, in any actor-critic framework.

To estimate the gradient from meta-rollouts of the tail tasks, we transform the integration of Eq. (6) into an expectation:

**Corollary 1.** Eq. (6) can be written as

$$\nabla_\theta J_\alpha^\theta(R) = \mathbb{E}_{\tau \sim D} \left[ \mathbb{E}_{\Lambda \sim P_\tau^\theta} [g(\Lambda)] \; \Big| \; V_\tau^\theta \le q_\alpha(V_\tau^\theta) \right], \tag{7}$$

where $g(\Lambda) = (R(\Lambda) - b) \sum_{\substack{1 \le k \le K \\ 1 \le t \le T}} \nabla_\theta \log \pi_\theta(a_{k,t}; \tilde{s}_{k,t})$; and $\tilde{s}_{k,t} = (s_{k,t}, h_{k,t})$ is the extended state (that includes all the task history $h_{k,t}$ until trajectory $k$, step $t$).

*Proof.* We apply the standard log trick $\nabla_\theta P_z^\theta = P_z^\theta \cdot \nabla_\theta \log P_z^\theta$ to Eq. (6), after substituting the meta-rollout PDF: $P_z^\theta(\Lambda) = \prod_{k=1}^{K} \left[ P_{0,z}(s_{k,0}) \cdot \prod_{t=1}^{T} P_z(s_{k,t+1}, r_{k,t} \,|\, s_{k,t}, a_{k,t}) \pi_\theta(a_{k,t}; \tilde{s}_{k,t}) \right]$. □

For a practical Monte-Carlo estimation of Eq. (7), given a task $z_i$, we need to estimate whether $V_{z_i}^\theta \le q_\alpha(V_\tau^\theta)$. To estimate $V_{z_i}^\theta$, we can generate $M$ i.i.d meta-rollouts with returns $\{R_{i,m}\}_{m=1}^{M}$, and calculate their average return $\hat{V}_{z_i}^\theta = R_i = \sum_{m=1}^{M} R_{i,m}/M$. Then, the quantile $q_\alpha(V_\tau^\theta)$ can be estimated over a batch of tasks $\hat{q}_\alpha = q_\alpha(\{R_i\}_{i=1}^{N})$. If $\hat{V}_{z_i}^\theta \le \hat{q}_\alpha$, we use *all* the meta-rollouts of $z_i$ for the gradient calculation (including meta-rollouts that by themselves have a higher return $R_{i,m} > \hat{q}_\alpha$). Notice that we use $M$ i.i.d meta-rollouts, each consisting of $K$ episodes (the episodes within a meta-rollout are *not* independent, due to agent memory).

Putting it together, we obtain the sample-based gradient estimator of Eq. (7):

$$\nabla_\theta J_\alpha^\theta(R) \approx \frac{1}{\alpha N} \sum_{i=1}^{N} \mathbf{1}_{R_i \le \hat{q}_\alpha^\theta} \sum_{m=1}^{M} g_{i,m},$$

$$g_{i,m} := (R_{i,m} - b) \sum_{k=1}^{K} \sum_{t=1}^{T} \nabla_\theta \log \pi_\theta(a_{i,m,k,t}; \tilde{s}_{i,m,k,t}), \tag{8}$$

where $a_{i,m,k,t}$, $\tilde{s}_{i,m,k,t}$ are the action and the state-and-history at task $i$, meta-rollout $m$, trajectory $k$ and step $t$.

The procedure described above follows the principles of CVaR-PG in (non-meta) RL, as the learning rule is only applied to the tail of the sampled batch. However, in MRL we consider a batch of tasks rather than a batch of trajectories. As discussed in Theorem 1 and its proof, this distinction has a substantial impact on the gradient and the resulting algorithm. Specifically, Eq. (8) allows for greater flexibility than Eq. (4), as it permits any baseline $b$ that does not depend on $\theta$. This allows gradient calculation using any PG algorithm, including SOTA methods such as PPO [Schulman et al., 2017] (which are already used in MRL methods such as VariBAD Zintgraf et al. [2019]). Therefore, in contrast to standard RL, CVaR optimization is not restricted to basic REINFORCE.

---

**Algorithm 1:** CVaR Meta Learning (CVaR-ML)

---

**1 Input**: Meta-learning algorithm (Definition 2); robustness level $\alpha \in (0, 1]$; task distribution $D$; $N$ tasks per batch; $M$ meta-rollouts per task

**2 while** not finished training **do**
  // Sample tasks
**3**  Sample $\{z_i\}_{i=1}^N \sim D$
  // Run meta-rollouts
**4**  $\{\{\Lambda_{i,m}\}_{m=1}^M\}_{i=1}^N \leftarrow \text{rollouts}(\{z_i\}_{i=1}^N, M)$
**5**  $R_{i,m} \leftarrow \text{return}(\Lambda_{i,m}), \quad \forall i, m$
  // Compute sample quantile
**6**  $R_i \leftarrow \text{mean}(\{R_{i,m}\}_{m=1}^M), \quad \forall i$
**7**  $\hat{q}_\alpha \leftarrow \text{quantile}(\{R_i\}_{i=1}^N, \alpha)$
  // Meta-learning algorithm train step
**8**  $\text{ML}\big(\{\Lambda_{i,m} \mid R_i \leq \hat{q}_\alpha, 1 \leq m \leq M\}\big)$

---

Our CVaR Meta Learning method (**CVaR-ML**, Algorithm 1) leverages this property to operate as a *meta-algorithm*, providing a robust version for any given baseline algorithm, such as Finn et al. [2017], Zintgraf et al. [2019]:

**Definition 2.** A *baseline MRL algorithm* learns a meta-policy $\pi_\theta$ using a training step ML. Given a batch of meta-rollouts $\{\Lambda_i\}$, $\text{ML}(\{\Lambda_i\})$ updates $\pi_\theta$.

Notice that CVaR-ML only handles task filtering, and uses the baseline training step ML as a black box (Line 8). Hence, it can be used with *any* MRL baseline – not just PG methods. In fact, by using a supervised meta-learning baseline, CVaR-ML can be applied to the supervised setting as well with minimal modifications, namely, replacing meta-rollouts with examples and returns with losses.

## 5 Efficient CVaR-ML

Theorem 1 guarantees unbiased gradients when using Algorithm 1; however, it does not bound their variance. In particular, Line 8 applies the learning step to a subset of only $\alpha NM$ meta-rollouts out of $NM$, which increases the estimator variance by a factor of $\alpha^{-1}$ compared to mean optimization. This could be prevented if we knew the set $\Omega_\alpha^\theta$ of tail-tasks (for the current $\pi_\theta$), and sampled only these tasks, using the distribution $D_\alpha^\theta(z) = \alpha^{-1}\mathbf{1}_{V_z^\theta \leq q_\alpha(V_\tau^\theta)}D(z)$. Proposition 1 shows that this would indeed recover the sample efficiency.

**Proposition 1** (Variance reduction). Denote by $G$ the estimator of $\nabla_\theta J_\alpha^\theta(R)$ in Eq. (8), assume there is no quantile error ($\hat{q}_\alpha^\theta = q_\alpha^\theta$), and denote $\mathbb{E}_D[\cdot] = \mathbb{E}_{z_i \sim D, R_{i,m} \sim P_{z_i}^\theta}[\cdot]$. Then, switching the task sample distribution to $D_\alpha^\theta$ leads to a variance reduction of factor $\alpha$:

$$\mathbb{E}_{D_\alpha^\theta}[\alpha G] = \mathbb{E}_D[G], \quad \text{Var}_{D_\alpha^\theta}(\alpha G) \leq \alpha \cdot \text{Var}_D(G).$$

*Proof sketch (the complete proof is in Appendix B).* We calculate the expectation and variance directly. $G$ is proportional to $\mathbf{1}_{R_i \leq q_\alpha^\theta}$ (Eq. (8)). The condition $R_i \leq q_\alpha^\theta$ leads to multiplication by the probability $\alpha$ when sampled from $D$ (where it corresponds to the $\alpha$-tail); but not when sampled from $D_\alpha^\theta$ (where it is satisfied w.p. 1). This factor $\alpha$ cancels out the ratio between the expectations of $G$ and $\alpha G$ (thus the expectations are identical) – but not the ratio $\alpha^2$ between their variances. □

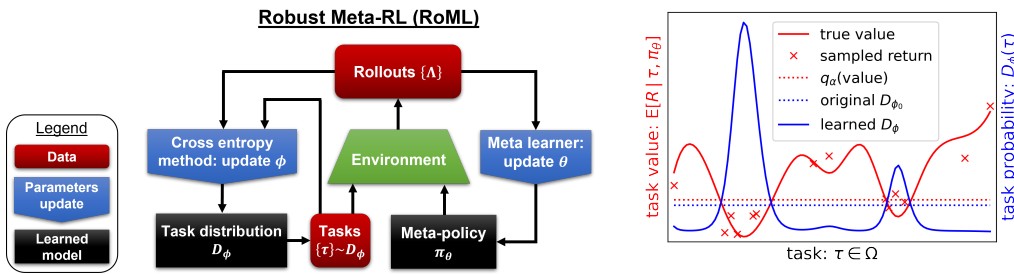

Figure 2: **Left**: RoML uses the cross entropy method to modify the task distribution $D_\phi$, which is used to generate the next meta-rollouts. **Right**: illustration of an arbitrary point of time in training: the task distribution $D_\phi$ (blue) is learned according to the task values of the current meta-policy $\pi_\theta$ (red). Since low-return tasks are over-sampled, the learned meta-policy is more robust to the selection of task.

Following the motivation of Proposition 1, we wish to increase the number of train tasks that come from the tail distribution $\Omega_\alpha^\theta$. To that end, we assume to have certain control over the sampling of training tasks. This assumption is satisfied in most simulated environments, as well as many real-world scenarios. For example, when training a driver, we choose the tasks, roads and times of driving throughout training. In this section, we propose a method to make these choices.

We begin with parameterizing the task distribution $D$: we consider a parametric family $D_\phi$ such that $D = D_{\phi_0}$. Then, we wish to modify the parameter $\phi$ so that $D_\phi$ aligns with $D_\alpha^\theta$ as closely as possible. To that end, we use the Cross Entropy Method (CEM, de Boer et al. [2005]), which searches for $\phi^*$ that minimizes the KL-divergence (i.e., cross entropy) between the two:

$$
\begin{aligned}
\phi^* \in \operatorname{argmin}_{\phi'} D_{KL}\left(D_\alpha^\theta \,\|\, D_{\phi'}\right) &= \operatorname{argmax}_{\phi'} \mathbb{E}_{z \sim D_{\phi_0}}\left[\mathbf{1}_{V_z^\theta \leq q_\alpha(V_\tau^\theta)} \log D_{\phi'}(z)\right] \\
&= \operatorname{argmax}_{\phi'} \mathbb{E}_{z \sim D_\phi}\left[w(z)\,\mathbf{1}_{V_z^\theta \leq q_\alpha(V_\tau^\theta)} \log D_{\phi'}(z)\right],
\end{aligned}
\tag{9}
$$

where $w(z) = \frac{D_{\phi_0}(z)}{D_\phi(z)}$ is the importance sampling weight corresponding to $z \sim D_\phi$. Note that Eq. (9) has a closed-form solution for most of the standard families of distributions [de Boer et al., 2005].

---

**Algorithm 2:** Robust Meta RL (RoML)

1 **Input**: Meta-learning algorithm (Definition 2); robustness level $\alpha \in (0, 1]$; parametric task distribution $D_\phi$; original parameter $\phi_0$; $N$ tasks per batch, $\nu \in [0, 1)$ of them sampled from the original $D_{\phi_0}$; CEM quantile $\beta \in (0, 1)$

2 **Initialize:**

3 $\quad \phi \leftarrow \phi_0, \quad N_o \leftarrow \lfloor \nu N \rfloor, \quad N_s \leftarrow \lceil (1 - \nu)N \rceil$

4 **while** not finished training **do**
   // Sample tasks

5 $\quad$ Sample $\{z_{o,i}\}_{i=1}^{N_o} \sim D_{\phi_0}, \quad \{z_{\phi,i}\}_{i=1}^{N_s} \sim D_\phi$

6 $\quad z \leftarrow (z_{o,1}, \dots, z_{o,N_o}, z_{\phi,1}, \dots, z_{\phi,N_s})$
   // Rollouts and meta-learning step

7 $\quad \{\Lambda_i\}_{i=1}^N \leftarrow \text{rollout}(\{z_i\}_{i=1}^N)$

8 $\quad R_i \leftarrow \text{return}(\Lambda_i), \quad \forall i \in \{1, \dots, N\}$

9 $\quad \text{ML}(\{\Lambda_i\}_{i=1}^N)$
   // Estimate reference quantile

10 $\quad w_{o,i} \leftarrow 1, \qquad\qquad \forall i \in \{1, \dots, N_o\}$

11 $\quad w_{\phi,i} \leftarrow \frac{D_{\phi_0}(z_{\phi,i})}{D_\phi(z_{\phi,i})}, \quad \forall i \in \{1, \dots, N_s\}$

12 $\quad w \leftarrow (w_{o,1}, \dots, w_{o,N_o}, w_{\phi,1}, \dots, w_{\phi,N_s})$

13 $\quad \hat{q}_\alpha \leftarrow \text{weighted\_quantile}(\{R_i\}, w, \alpha)$
   // Compute sample quantile

14 $\quad q_\beta \leftarrow \text{quantile}(\{R_i\}, \beta)$
   // Update sampler

15 $\quad q \leftarrow \max(\hat{q}_\alpha, q_\beta)$

16 $\quad \phi \leftarrow \operatorname{argmax}_{\phi'} \sum_{i \leq N} w_i \, \mathbf{1}_{R_i \leq q} \log D_{\phi'}(z_i)$

---

For a batch of $N$ tasks sampled from $D = D_{\phi_0}$, Eq. (9) essentially chooses the $\alpha N$ tasks with the lowest returns, and updates $\phi$ to focus on these tasks. This may be noisy unless $\alpha N \gg 1$. Instead, the CEM chooses a larger number of tasks $\beta > \alpha$ for the update, where $\beta$ is a hyper-parameter. $\phi$ is updated according to these $\beta N$ lowest-return tasks, and the next batch is sampled from $D_\phi \neq D_{\phi_0}$. This repeats iteratively: every batch is sampled from $D_\phi$, where $\phi$ is updated according to the $\beta N$ lowest-return tasks of the former batch. Each task return is also compared to the $\alpha$-quantile of the *original* distribution $D_{\phi_0}$. If more than $\beta N$ tasks yield lower returns, the CEM permits more samples for the update step. The return quantile over $D_{\phi_0}$ can be estimated from $D_\phi$ at any point using importance sampling weights. See more details about the CEM in Appendix C.1.

In our problem, the target distribution is the tail of $D_{\phi_0}$. Since the tail is defined by the agent returns in these tasks, it varies with the agent and is non-stationary throughout training. Thus, we use the dynamic-target CEM of Greenberg [2022]. To smooth the changes in the sampled tasks, the sampler is also regularized to always provide certain exposure to all the tasks: we force $\nu$ percent of every batch to be sampled from the original distribution $D = D_{\phi_0}$, and only $1 - \nu$ percent from $D_\phi$.

Putting this together, we obtain the Robust Meta RL algorithm (**RoML**), summarized in Algorithm 2 and Fig. 2. RoML does not require multiple meta-rollouts per update (parameter $M$ in Algorithm 1), since it directly models high-risk tasks. Similarly to CVaR-ML, RoML is a meta-algorithm and can operate on top of any meta-learning baseline (Definition 2). Given the baseline implementation, only the tasks sampling procedure needs to be modified, which makes RoML easy to implement.

**Limitations:** The CEM's adversarial tasks sampling relies on several assumptions. Future research may reduce some of these assumptions, while keeping the increased data efficiency of RoML.

First, as mentioned above, we need at least partial control over the selection of training tasks. This assumption is common in other RL frameworks [Dennis et al., 2020, Jiang et al., 2021], and often holds in both simulations and the real world (e.g., choosing in which roads and hours to train driving).

Second, the underlying task distribution $D$ is assumed to be known, and the sample distribution is limited to the chosen parameterized family $\{D_\phi\}$. For example, if $\tau \sim U([0,1])$, the user may choose the family of Beta distributions $Beta(\phi_1, \phi_2)$ (where $Beta(1,1) \equiv U([0,1])$), as demonstrated in Appendix D.2. The selected family expresses implicit assumptions on the task-space. For example, if the probability density function is smooth, close tasks will always have similar sample probabilities; and if the family is unimodal, high-risk tasks can only be over-sampled from around a single peak. This approach is useful for generalization across continuous task-spaces – where the CEM can never observe the infinitely many possible tasks. Yet, it may pose limitations in certain discrete task-spaces, if there is no structured relationship between tasks.

## 6 Experiments

We implement **RoML** and **CVaR-ML** on top of two different risk-neutral MRL baselines – **VariBAD** [Zintgraf et al., 2019] and **PEARL** [Rakelly et al., 2019]. As a risk-averse reference for comparison, we use **CeSoR** [Greenberg et al., 2022], an efficient sampling-based method for CVaR optimization in RL, implemented on top of PPO. As another reference, we use the Unsupervised Environment Design algorithm **PAIRED** [Dennis et al., 2020], which uses regret minimization to learn robust policies on a diverse set of tasks.

Section 6.1 demonstrates the mean/CVaR tradeoff, as our methods learn substantially different policies from their baseline. Section 6.2 demonstrates the difficulty of the naive CVaR-ML in more challenging control benchmarks, and the RoML's efficacy in them. The ablation test in Appendix D.4 demonstrates that RoML deteriorates significantly when the CEM is replaced by a naive adversarial sampler. Section 6.3 presents an implementation of CVaR-ML and RoML on top of **MAML** [Finn et al., 2017] for *supervised* meta-learning. In all the experiments, the running times of RoML and CVaR-ML are indistinguishable from their baselines (RoML's CEM computations are negligible).

**Hyper-parameters:** To test the practical applicability of RoML as a meta-algorithm, in every experiment, **we use the same hyper-parameters for RoML, CVaR-ML and their baseline**. In particular, we use the baseline's default hyper-parameters whenever applicable (Zintgraf et al. [2019], Rakelly et al. [2019] in Section 6.2, and Finn et al. [2017] in Section 6.3). That is, we use the same hyper-parameters as originally tuned for the baseline, and test whether RoML improves the robustness without any further tuning of them. As for the additional hyper-parameters of the meta-algorithm itself: in Algorithm 1, we use $M = 1$ meta-rollout per task; and in Algorithm 2, we use $\beta = 0.2, \nu = 0$ unless specified otherwise (similarly to the CEM in Greenberg et al. [2022]). For the references PAIRED and CeSoR, we use the hyper-parameters of Dennis et al. [2020], Greenberg et al. [2022]. Each experiment is repeated for 30 seeds. See more details in Appendix D. The code is available in our repositories: VariBAD, PEARL, CeSoR, PAIRED and MAML.

### 6.1 Khazad Dum

We demonstrate the tradeoff between mean and CVaR optimization in the Khazad Dum benchmark, visualized in Fig. 3. The agent begins at a random point in the bottom-left part of the map, and has to reach the green target as quickly as possible, without falling into the black abyss. The bridge is not covered and thus is exposed to wind and rain, rendering its floor slippery and creating an additive action noise (Fig. 3b) – to a level that varies with the weather. Each task is characterized by the rain intensity, which is exponentially distributed. The CEM in RoML is allowed to modify the parameter of this exponential distribution. Note that the agent is not aware of the current task (i.e, the weather), but may infer it from observations. We set the target risk level to $\alpha = 0.01$, and train each meta-agent for a total of $5 \cdot 10^6$ frames. See complete details in Appendix D.1.

We implement CVaR-ML and RoML on top of VariBAD. As shown in Fig. 3, VariBAD learns to try the short path – at the risk of rare falls into the abyss. By contrast, our CVaR-optimizing methods take the longer path and avoid risk. This policy increases the cumulative cost of time-steps, but leads to higher CVaR returns, as shown in Fig. 3f. In addition to superior CVaR, RoML also provides competitive *average* returns in this example (Fig. 3e). Finally, in accordance with Proposition 1, RoML learns significantly faster than CVaR-ML.

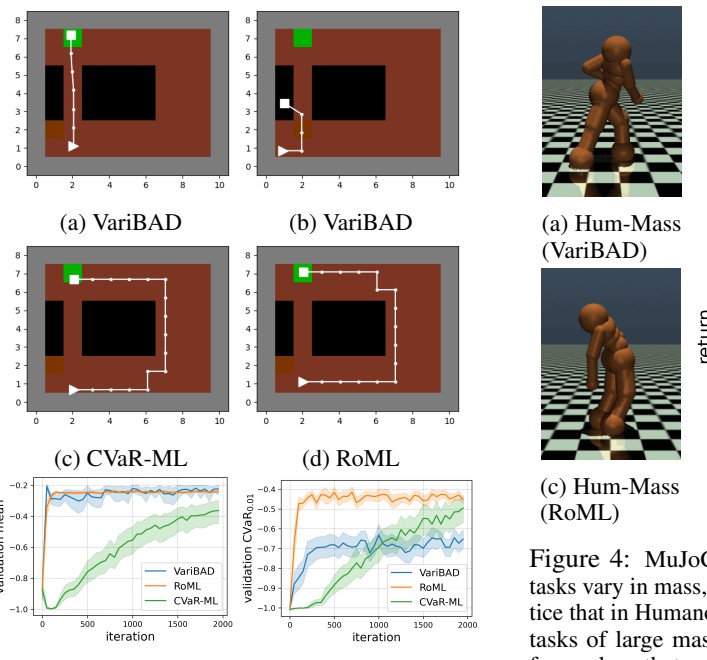

(a) VariBAD  (b) VariBAD

(c) CVaR-ML  (d) RoML

(e) Mean  (f) CVaR

Figure 3: Khazad-Dum: (a-d) Sample episodes. (e-f) Test return vs. training iteration, with 95% confidence intervals over 30 seeds.

(a) Hum-Mass (VariBAD)  (b) HalfCheetah-Body

(c) Hum-Mass (RoML)  (d) HalfCheetah-Mass

Figure 4: MuJoCo: (a-c) Sample frames, where tasks vary in mass, head size and damping level. Notice that in Humanoid, RoML handles the low-return tasks of large mass by leaning the center of mass forward, so that gravity pulls the humanoid forward. (d) Average return per range of tasks in HalfCheetah-Mass. RoML learns to act robustly: it is less sensitive to the task, and in particular performs better on high-risk tasks.

## 6.2 Continuous Control

We rely on standard continuous control problems from the MuJoCo framework [Todorov et al., 2012]: training a cheetah to run (**HalfCheetah**), and training a **Humanoid** and an **Ant** to walk. For each of the 3 environments, we create 3 meta-learning versions: (1) *Goal* or *Vel* [Finn et al., 2017], where each task corresponds to a different location or velocity objective, respectively; (2) *Mass*, where each task corresponds to a different body mass; and (3) *Body*, where each task corresponds to different mass, head size and physical damping level (similarly to Wang and Van Hoof [2022]). In addition, to experiment with high-dimensional task spaces, we randomly draw 10 numeric variables from *env.model* in HalfCheetah, and let them vary between tasks. We define 3 such environments with different random sets of task variables (**HalfCheetah 10D-task a,b,c**). For each of the 12 environments above, we set a target risk level of $\alpha = 0.05$ and optimize for $K = 2$ episodes per task. Additional implementation details are specified in Appendix D.2.

Interestingly, the naive approach of **CVaR-ML** consistently fails to meta-learn in all the cheetah environments. It remains unsuccessful even after large number of steps, indicating a difficulty beyond sample inefficiency. A possible explanation is the effectively decreased batch size of CVaR-ML. **PAIRED** and **CeSoR** also fail to adjust to the MRL environments, and obtain poor CVaR returns.

**RoML**, on the other hand, consistently improves the CVaR returns (Table 1) compared to its baseline (VariBAD or PEARL), while using the same hyper-parameters as the baseline. The VariBAD baseline presents better returns and running times than PEARL on HalfCheetah, and thus is used for the 6 extended environments (Humanoid and Ant). RoML improves the CVaR return in comparison to the baseline algorithm in all the 18 experiments (6 with PEARL and 12 with VariBAD).

In 5 out of 18 experiments, RoML slightly improves the *average* return compared to its baseline, and not only the CVaR (Table 2 in the appendix). This indicates that low-return tasks can sometimes be improved at the cost of high-return tasks, but without hurting average performance. In addition, this may indicate that over-sampling difficult tasks forms a helpful learning curriculum.

The robustness of RoML to the selected task is demonstrated in Fig. 4d. In multi-dimensional task spaces, RoML learns to focus the sampling modification on the high-impact variables, as demonstrated

Table 1: $CVaR_{0.05}$ return over 1000 test tasks, for different models and MuJoCo environments. Standard deviation is presented over 30 seeds. Mean returns are displayed in Table 2.

| | HalfCheetah | | | HalfCheetah 10D-task | | |
| --- | --- | --- | --- | --- | --- | --- |
| | Vel | Mass | Body | (a) | (b) | (c) |
| CeSoR | $-2606 \pm 25$ | $902 \pm 36$ | $478 \pm 27$ | $637 \pm 26$ | $981 \pm 31$ | $664 \pm 26$ |
| PAIRED | $-725 \pm 65$ | $438 \pm 37$ | $218 \pm 51$ | $229 \pm 59$ | $354 \pm 53$ | $81 \pm 65$ |
| CVaR-ML | $-897 \pm 23$ | $38 \pm 6$ | $76 \pm 5$ | $120 \pm 11$ | $141 \pm 11$ | $81 \pm 4$ |
| PEARL | $-1156 \pm 23$ | $1115 \pm 19$ | $800 \pm 5$ | $1140 \pm 33$ | $1623 \pm 23$ | $\mathbf{1016 \pm 5}$ |
| VariBAD | $-202 \pm 6$ | $1072 \pm 16$ | $835 \pm 30$ | $1126 \pm 6$ | $1536 \pm 39$ | $988 \pm 13$ |
| RoML (VariBAD) | $\mathbf{-184 \pm 4}$ | $\mathbf{1259 \pm 19}$ | $\mathbf{935 \pm 17}$ | $\mathbf{1227 \pm 13}$ | $\mathbf{1697 \pm 24}$ | $999 \pm 20$ |
| RoML (PEARL) | $-1089 \pm 31$ | $1186 \pm 34$ | $808 \pm 6$ | $1141 \pm 27$ | $1657 \pm 18$ | $1024 \pm 6$ |

| | Humanoid | | | Ant | | |
| --- | --- | --- | --- | --- | --- | --- |
| | Vel | Mass | Body | Goal | Mass | Body |
| VariBAD | $801 \pm 10$ | $1283 \pm 18$ | $1290 \pm 19$ | $-500 \pm 9$ | $1370 \pm 6$ | $\mathbf{1365 \pm 4}$ |
| RoML (VariBAD) | $\mathbf{833 \pm 4}$ | $\mathbf{1378 \pm 20}$ | $\mathbf{1365 \pm 21}$ | $\mathbf{-454 \pm 8}$ | $\mathbf{1385 \pm 3}$ | $\mathbf{1368 \pm 4}$ |

in Fig. 11 in the appendix. Finally, qualitative inspection shows that RoML learns to handle larger masses, for example, by leaning forward and letting gravity pull the agent forward, as displayed in Fig. 4c (see animations on GitHub).

### 6.3 Beyond RL: Robust Supervised Meta-Learning

Our work focuses on robustness in MRL. However, the concept of training on harder data to improve robustness, as embodied in RoML, is applicable beyond the scope of RL. As a preliminary proof-of-concept, we apply RoML to a toy supervised meta-learning problem of sine regression, based on Finn et al. [2017]: The input is $x \in [0, 2\pi]$, the desired output is $y = A\sin(\omega x + b)$, and the task is defined by the parameters $\tau = (A, b, \omega)$. Similarly to Finn et al. [2017], the model is fine-tuned for each task via a gradient-descent optimization step over 10 samples $\{(x_i, y_i)\}_{i=1}^{10}$, and is tested on another set of 10 samples. The goal is to find model weights that adapt quickly to new task data.

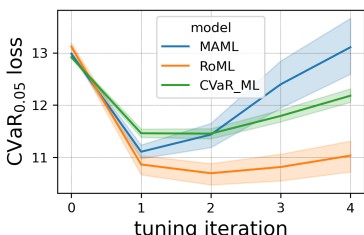

Figure 5: Supervised Sine Regression: CVaR loss over 10000 test tasks, against the number of tuning gradient-steps at test time. The 95% confidence intervals are calculated over 30 seeds.

We implement CVaR-ML and RoML on top of MAML [Finn et al., 2017]. As shown in Fig. 5, RoML achieves better CVaR losses over tasks than both CVaR-ML and MAML. The complete setting and results are presented in Appendix E.

## 7 Summary and Future Work

We defined a robust MRL objective and derived the CVaR-ML algorithm to optimize it. In contrast to its analogous algorithm in standard RL, we proved that CVaR-ML does not present biased gradients, yet it does inherit the same data inefficiency. To address the latter, we introduced RoML and demonstrated its advantage in sample efficiency and CVaR return.

Future research may address the CEM-related limitations of RoML discussed at the end of Section 5. Another direction for future work is extension of RoML to other scenarios, especially where a natural task structure can be leveraged to improve task robustness, e.g., in supervised learning and coordinate-based regression [Tancik et al., 2020].

Finally, RoML is easy to implement, operates agnostically as a meta-algorithm on top of existing MRL methods, and can be set to any desired robustness level. We believe that these properties, along with our empirical results, make RoML a promising candidate for MRL in risk-sensitive applications.

**Acknowledgements:** This work has received funding from the European Union's Horizon Europe Programme, under grant number 101070568.

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

# Contents

# A    Policy Gradient for CVaR Optimization in Meta RL

In this section we provide the complete proof for Theorem 1. For completeness, Appendix A.1 recaps of the analogous proof in standard (non-meta) RL, before moving on to the proof in Appendix A.2. This allows us to highlight the differences between the two.

The substantial difference is that in RL, the CVaR is defined directly over the low-return trajectories, and the policy parameter $\theta$ affects the probability of each trajectory in the tail (Eq. (10)). In MRL (Eq. (2)), on the other hand, the CVaR is defined over the low-return tasks, whose probability is not affected directly by $\theta$ (Eq. (12)). This allows a decoupling between $\theta$ and $\tau$, which results in Theorem 1.

Another high-level intuition is as follows. In RL, the CVaR-PG gradient is invariant to successful strategies, hence must be explicitly negative for the unsuccessful ones (in order not to encourage them, see Fig. 6). In MRL, within the tasks of interest, the gradient always encourages successful strategies on account of the unsuccessful ones (Fig. 7).

## A.1    Recap: PG for CVaR in (non-meta) RL

We briefly recap the calculation of Propositions 1 and 2 in Tamar et al. [2015] for CVaR policy gradient under the standard RL settings.

**Definition 3** (CVaR return in (non-meta) RL). Consider an MDP $(S, A, P, \gamma, P_0)$ with the cumulative reward (i.e., return) $R \sim P^\theta$, whose $\alpha$-quantile is $q_\alpha^\theta(R)$. Recall the CVaR objective defined in Section 3:

$$\tilde{J}_\alpha^\theta(R) = \text{CVaR}_{R \sim P^\theta}^\alpha[R] = \int_{-\infty}^{q_\alpha^\theta(R)} x \cdot P^\theta(x) \cdot dx$$

To calculate the policy gradient of $\tilde{J}_\alpha^\theta(R)$, we begin with the conservation of probability mass below the quantile $q_\alpha^\theta$:

$$\int_{-\infty}^{q_\alpha^\theta(R)} P^\theta(x)dx \equiv \alpha.$$

Then, using the Leibniz integral rule, we have

$$0 = \nabla_\theta \int_{-\infty}^{q_\alpha^\theta(R)} P^\theta(x)dx = \left[ \int_{-\infty}^{q_\alpha^\theta(R)} \nabla_\theta P^\theta(x)dx \right] + \left[ P^\theta(q_\alpha^\theta(R)) \cdot \nabla_\theta q_\alpha^\theta(R) \right]. \tag{10}$$

Notice that as a particular consequence of the conservation rule Eq. (10), positive gradients of $P^\theta(x)$ cause the quantile $q_\alpha^\theta(R)$ to decrease. This phenomenon makes the CVaR policy gradient sensitive to the selection of baseline, as visualized in Fig. 6b. In fact, the quantile $q_\alpha^\theta(R)$ itself is the only baseline that permits unbiased gradients:

$$\nabla_\theta \tilde{J}_\alpha^\theta(R) = \nabla_\theta \int_{-\infty}^{q_\alpha^\theta(R)} x \cdot P^\theta(x) \cdot dx =$$

$$\left[ \int_{-\infty}^{q_\alpha^\theta(R)} x \cdot \nabla_\theta P^\theta(x)dx \right] + \left[ q_\alpha^\theta(R) \cdot P^\theta(q_\alpha^\theta(R)) \cdot \nabla_\theta q_\alpha^\theta(R) \right] \overset{Eq.\ (10)}{\overbrace{=}}$$

$$\left[ \int_{-\infty}^{q_\alpha^\theta(R)} x \cdot \nabla_\theta P^\theta(x)dx \right] - \left[ q_\alpha^\theta(R) \cdot \int_{-\infty}^{q_\alpha^\theta(R)} \nabla_\theta P^\theta(x)dx \right] = \tag{11}$$

$$\int_{-\infty}^{q_\alpha^\theta(R)} (x - q_\alpha^\theta(R)) \cdot \nabla_\theta P^\theta(x)dx,$$

which gives us Eq. (4).

## A.2    PG for CVaR in Meta RL

We now turn to PG for CVaR optimization in *MRL*. We rely on the definitions and notations of Section 3, as well as Definition 1 and Assumption 1 in Section 4. Notice that $J_\alpha^\theta(R)$ of Eq. (2) can be

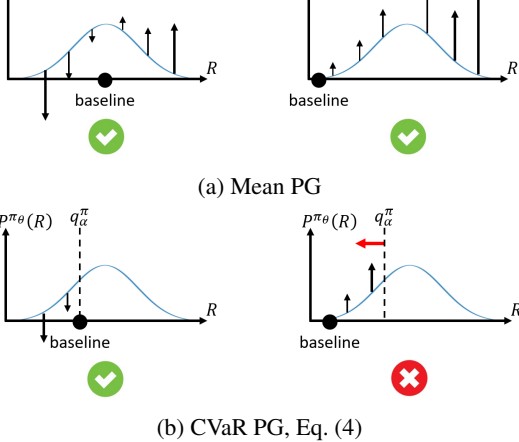

(a) Mean PG

(b) CVaR PG, Eq. (4)

Figure 6: Illustration of the policy gradient estimation in standard RL. (a) In Mean PG, the expected gradient is independent of the baseline: even if most of the distribution $P^{\pi_\theta}$ seems to be "pushed upwards", its normalization to a total probability of 1 forces the probability of low returns to decrease for that of high returns will increase. (b) In CVaR PG, due to the effect of Eq. (10), any baseline except for $q_\alpha^\theta(R)$ leads to biased gradients.

written in integral form as

$$J_\alpha^\theta(R) = \texttt{CVaR}_{\tau \sim D}^\alpha \left[ \mathbb{E}_{R \sim P_\tau^\theta}[R] \right] = \int_{\Omega_\alpha^\theta} D(z) \int_{-\infty}^\infty x \cdot P_z^\theta(x) \cdot dx \cdot dz.$$

In Eq. (11) above, the boundary of the integral over the $\alpha$-tail is simply the scalar $q_\alpha^\theta$. In MRL, this is replaced by the boundary of the set $\Omega_\alpha^\theta$, defined in a general topological space. Thus, we begin by characterizing this boundary.

**Lemma 1** (The boundary of $\Omega_\alpha^\theta$). Under Assumption 1, $\forall z \in \partial \Omega_\alpha^\theta : \int_{-\infty}^\infty x \cdot P_z^\theta(x) \cdot dx = q_\alpha(V_\tau^\theta)$.

*Proof.* Since $v(z) = V_z^\theta$ is a continuous function between topological spaces, by denoting $B = (-\infty, q_\alpha(V_\tau^\theta)]$ we have

$$\partial \Omega_\alpha^\theta = \partial v^{-1}(B) \overset{\overbrace{\text{continuous } v}}{\subseteq} v^{-1}(\partial B) = v^{-1}(\{q_\alpha(V_\tau^\theta)\}) = \{z \in \Omega \,|\, V_z^\theta = q_\alpha(V_\tau^\theta)\},$$

hence $\forall z \in \partial \Omega_\alpha^\theta : V_z^\theta = q_\alpha(V_\tau^\theta)$. Notice that $V_z^\theta = \int_{-\infty}^\infty x P_z^\theta(x) \cdot dx$. $\qquad \square$

Finally, we can prove Theorem 1.

*Proof of Theorem 1.* First, we consider the conservation of probability:

$$\int_{\Omega_\alpha^\theta} D(z) dz \equiv \alpha$$

The gradient of this integral can be calculated using a high-dimensional generalization of the Leibniz integral rule, named Reynolds Transport Theorem (RTT, Tromba and Marsden [1996]):

$$0 \overset{\overbrace{\substack{\text{derivative of} \\ \text{a constant}}}}{=} \nabla_\theta \int_{\Omega_\alpha^\theta} D(z) dz \overset{\overbrace{\text{RTT}}}{=} \int_{\Omega_\alpha^\theta} \nabla_\theta D(z) dz + \int_{\partial \Omega_\alpha^\theta} D(z) \cdot (v_b \cdot n) \cdot dA$$

$$\overset{\overbrace{\nabla_\theta D(z) \equiv 0}}{=} \int_{\partial \Omega_\alpha^\theta} D(z) \cdot (v_b \cdot n) \cdot dA, \tag{12}$$

where $n(z, \theta)$ is the outward-pointing unit vector that is normal to the surface $\partial \Omega_\alpha^\theta$, and $v_b(z, \theta)$ is the velocity of the area element on the surface with area $dA$. Notice that $\nabla_\theta D(z) \equiv 0$ in the last

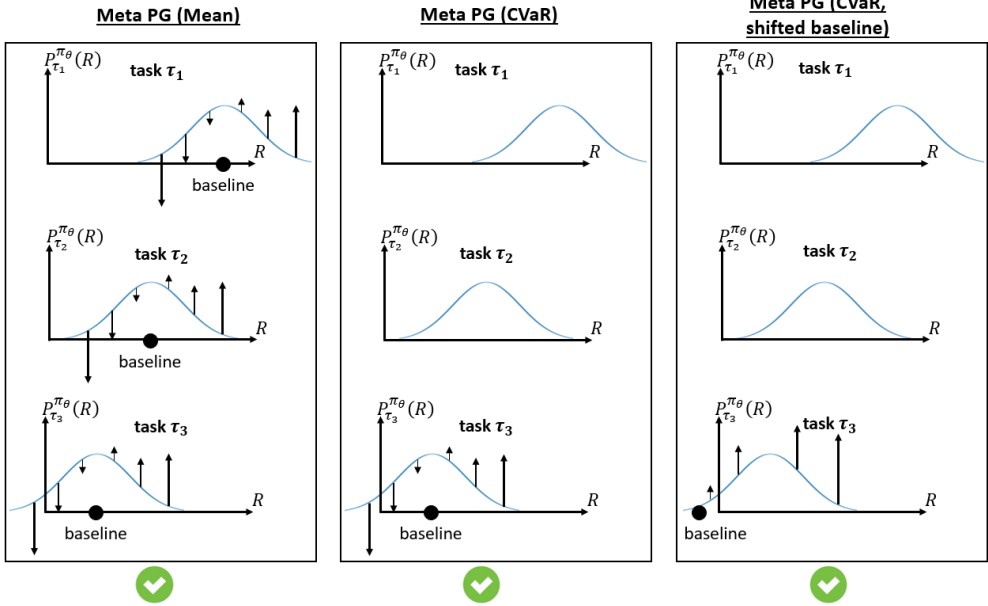

Figure 7: Illustration of the meta policy gradient estimation. In Mean meta-PG (left, Eq. (3)), the PG is estimated for all tasks. In CVaR meta-PG (center, Eq. (6)), it is only calculated over the low-return tasks. Since the returns distribution is decoupled from the tasks distribution, shifting the baseline (right) does not insert bias to the PG, in accordance with Theorem 1.

equality expresses a substantial difference from the standard RL settings in Eq. (10): there, we have $\nabla_\theta P^\theta(x)$, which does not necessarily vanish.

Next, we turn to the meta policy gradient itself, again using Reynolds Transport Theorem:

$$
\nabla_\theta J_\alpha^\theta(R) = \nabla_\theta \int_{\Omega_\alpha^\theta} D(z) \int_{-\infty}^{\infty} x P_z^\theta(x) \cdot dx \cdot dz \overset{RTT}{=}
$$

$$
\left[ \int_{\Omega_\alpha^\theta} D(z) \int_{-\infty}^{\infty} x \nabla_\theta P_z^\theta(x) \cdot dx \cdot dz \right] + \left[ \int_{\partial\Omega_\alpha^\theta} D(z) \left( \int_{-\infty}^{\infty} x P_z^\theta(x) \cdot dx \right) \cdot (v_b \cdot n) \cdot dA \right] \overset{Lemma\ 1}{=}
$$

$$
\left[ \int_{\Omega_\alpha^\theta} D(z) \int_{-\infty}^{\infty} x \nabla_\theta P_z^\theta(x) \cdot dx \cdot dz \right] + \left[ q_\alpha(V_\tau^\theta) \int_{\partial\Omega_\alpha^\theta} D(z) \cdot (v_b \cdot n) \cdot dA \right] \overset{Eq.\ (12)}{=}
$$

$$
\int_{\Omega_\alpha^\theta} D(z) \int_{-\infty}^{\infty} x \nabla_\theta P_z^\theta(x) \cdot dx \cdot dz.
$$

$$(13)$$

Finally, we show that any $\theta$-independent additive baseline (highlighted in the equation) does not change the gradient calculation:

$$\int_{\Omega_\alpha^\theta} D(z) \int_{-\infty}^\infty (x-b)\nabla_\theta P_z^\theta(x) \cdot dx \cdot dz =$$

$$\left[\int_{\Omega_\alpha^\theta} D(z) \int_{-\infty}^\infty x\nabla_\theta P_z^\theta(x) \cdot dx \cdot dz\right] - \left[\int_{\Omega_\alpha^\theta} D(z) \int_{-\infty}^\infty b\nabla_\theta P_z^\theta(x) \cdot dx \cdot dz\right] =$$

$$\left[\int_{\Omega_\alpha^\theta} D(z) \int_{-\infty}^\infty x\nabla_\theta P_z^\theta(x) \cdot dx \cdot dz\right] - \left[\int_{\Omega_\alpha^\theta} D(z) \cdot b \cdot \nabla_\theta \left(\int_{-\infty}^\infty P_z^\theta(x) \cdot dx\right) \cdot dz\right] \overset{\int_{-\infty}^\infty P_z^\theta(x)dx \equiv 1}{=}$$

$$\left[\int_{\Omega_\alpha^\theta} D(z) \int_{-\infty}^\infty x\nabla_\theta P_z^\theta(x) \cdot dx \cdot dz\right] - 0 = \nabla_\theta J_\alpha^\theta(R),$$

(14)

which completes the proof. Notice that we used the identity $\nabla_\theta \int_{-\infty}^\infty P_z^\theta(x)dx = \nabla_\theta 1 = 0$; this does not hold for the analogous term in the standard RL settings in Eq. (10), $\nabla_\theta \int_{-\infty}^{q_\alpha^\theta(x)} P^\theta(x)dx$, whose gradient depends on $\nabla_\theta q_\alpha^\theta(x)$ according to the Leibniz integral rule.

$\square$

## B  Proof of Proposition 1

*Proof.* Recall that by Eq. (8), $G = \frac{1}{\alpha N} \sum_{i=1}^N \mathbf{1}_{R_i \leq \hat{q}_\alpha^\theta} \sum_{m=1}^M g_{i,m}$. Denoting $G_i = \sum_{m=1}^M g_{i,m}$ and substituting $\hat{q}_\alpha^\theta = q_\alpha^\theta$, we have

$$G = \frac{1}{N} \sum_{i=1}^N \alpha^{-1} \mathbf{1}_{R_i \leq q_\alpha^\theta} G_i.$$

**Expectation:**  Since $\{G_i\}$ are i.i.d, and using the law of total probability, we obtain

$$\mathbb{E}_D[\alpha^{-1}\mathbf{1}_{R_i \leq q_\alpha^\theta} G_i] = \alpha \cdot \left(\alpha^{-1} \cdot 1 \cdot \mathbb{E}_D[G_1 \mid R_1 \leq q_\alpha^\theta]\right) + (1-\alpha) \cdot \left(\alpha^{-1} \cdot 0 \cdot \mathbb{E}_D[G_1 \mid R_1 > q_\alpha^\theta]\right)$$
$$= \mathbb{E}_D[G_1 \mid R_1 \leq q_\alpha^\theta].$$

By switching the task sample distribution to $D_\alpha^\theta$, and using the definition of $D_\alpha^\theta$, we simply have

$$\mathbb{E}_{D_\alpha^\theta}[\alpha^{-1}\mathbf{1}_{R_i \leq q_\alpha^\theta} G_i] = \alpha^{-1}\mathbb{E}_{D_\alpha^\theta}[G_1] = \alpha^{-1}\mathbb{E}_D[G_1 \mid R_1 \leq q_\alpha^\theta].$$

Together, we obtain $\mathbb{E}_{D_\alpha^\theta}[\alpha G] = \mathbb{E}_D[G]$ as required.

**Variance:**  For the original distribution, since $\{G_i\}$ are i.i.d, we have

$$N \cdot \mathrm{Var}_D(G) = \mathrm{Var}_D(\alpha^{-1}\mathbf{1}_{R_1 \leq q_\alpha^\theta} G_1)$$
$$= \mathbb{E}_D[\alpha^{-2}\mathbf{1}_{R_1 \leq q_\alpha^\theta} G_1^2] - \mathbb{E}_D[\alpha^{-1}\mathbf{1}_{R_1 \leq q_\alpha^\theta} G_1]^2$$
$$= \mathbb{E}_D[\alpha\alpha^{-2}G_1^2 \mid R_1 \leq q_\alpha^\theta] - \mathbb{E}_D[\alpha\alpha^{-1}G_1 \mid R_1 \leq q_\alpha^\theta]^2$$
$$= \alpha^{-1}\mathbb{E}_{D_\alpha^\theta}[G_1^2] - \mathbb{E}_{D_\alpha^\theta}[G_1]^2$$
$$\geq \alpha^{-1}(\mathbb{E}_{D_\alpha^\theta}[G_1^2] - \mathbb{E}_{D_\alpha^\theta}[G_1]^2)$$
$$= \alpha^{-1}\mathrm{Var}_{D_\alpha^\theta}(G_1).$$

For the tail distribution $D_\alpha^\theta$, however,

$$N \cdot \mathrm{Var}_{D_\alpha^\theta}(\alpha G) = \alpha^2 \mathrm{Var}_{D_\alpha^\theta}(\alpha^{-1}\mathbf{1}_{R_1 \leq q_\alpha^\theta} G_1) = \mathrm{Var}_{D_\alpha^\theta}(G_1),$$

which completes the proof.

$\square$

## C The Cross Entropy Method

### C.1 Background

The Cross Entropy Method (CEM, de Boer et al. [2005]) is a general approach to rare-event sampling and optimization. In this work, we use its sampling version to sample high-risk tasks from the tail of $D$. As described in Section 5, the CEM repeatedly samples from the parameterized distribution $D_\phi$, and updates $\phi$ according to the $\beta$-tail of the sampled batch. Since every iteration focuses on the tail of its former, we intuitively expect exponential convergence to the tail of the original distribution. While theoretical convergence analysis does not guarantee the exponential rate [de Mello and Rubinstein, 2003], practically, the CEM often converges within several iterations. For clarity, we provide the pseudo-code for the basic CEM in Algorithm 3. In this version, the CEM repeatedly generates samples from the tail of the given distribution $D_{\phi_0}$.

---

**Algorithm 3:** The Cross Entropy Method (CEM)

1 **Input**: distribution $D_{\phi_0}$; score function $R$; target level $q$; batch size $N$; CEM quantile $\beta$.

2 $\phi \leftarrow \phi_0$
3 **while** true **do**
   // Sample
4   Sample $z \sim D_\phi^N$
5   $w_i \leftarrow D_{\phi_0}(z_i)/D_\phi(z_i) \quad (1 \leq i \leq N)$
6   Print $z$
   // Update
7   $q' \leftarrow \max\left(q,\ q_\beta\left(\{R(z_i)\}_{i=1}^N\right)\right)$
8   $\phi \leftarrow \mathrm{argmax}_{\phi'} \sum_{i=1}^N w_i \mathbf{1}_{R(z_i) \leq q'} \log D_{\phi'}(z_i)$

---

### C.2 Discussion

The CEM is the key to the flexible robustness level of RoML (Algorithm 2): it can learn to sample not only the single worst-case task, but all the $\alpha$ tasks with the lowest returns.

The CEM searches for a task distribution within a parametric family of distributions. This approach can handle infinite task spaces, and learn the difficulty of tasks in the entire task space from a mere finite sample of tasks. For example, assume that the tasks correspond to environment parameters that take continuous values within some bounded box (as in Section 6.2 and Section 6.3). The CEM can fit a distribution over an entire subset of the box – from a mere finite batch of tasks. This property lets the CEM learn the high-risk tasks quickly and accelerates the meta-training, as demonstrated in Section 6 and Appendix D.3.

On the other hand, this approach relies on the structure in the task space. If the tasks do not have a natural structure like the ones in the bounded box, it is not trivial to define the parametric family of distributions. This is the case in certain supervised meta learning problems. For example, in the common meta-classification problem [Finn et al., 2017], the task space consists of subsets of classes to classify. This is a discrete space without a trivial metric between tasks. Hence, it is difficult for the CEM to learn the $\alpha$ lowest-return tasks from a finite sample. Thus, while RoML is applicable to supervised meta learning as well as MRL, certain task spaces require further adjustment, such as a meaningful embedding of the task space. This challenge is left for future work.

## D Experiments: Detailed Settings and Results

### D.1 Khazad Dum

> *At the end of the hall the floor vanished and fell to an unknown depth. The outer door could only be reached by a slender bridge of stone, without kerb or rail, that spanned the chasm with one curving spring of fifty feet. It was an ancient defence of the Dwarves against any enemy that might capture the First Hall and the outer passages. They could only pass across it in single file.* [Tolkien, 1954]

The detailed settings of the Khazad Dum environment presented in Section 6.1 are as follows. Every task is carried for $K = 4$ episodes of $T = 32$ times steps. The return corresponds to the undiscounted sum of the rewards ($\gamma = 1$). Every time step has a cost of $1/T$ points if the L1-distance of the agent from the target is larger than 5; and for distances between 0 and 5, the cost varies linearly between 0 and $1/T$. By reaching the destination, the agent obtains a reward of $5/T$, and has no more costs for the rest of the episode. By falling to the abyss, the agent can no longer reach the goal and is bound to suffer a cost of $1/T$ for every step until the end of the episode. Every step, the agent observes its location (represented using a soft one-hot encoding, similarly to Greenberg et al. [2022]) and chooses whether to move left, right, up or down. If the agent attempts to move into a wall, it remains in place.

The tasks are characterized by the rain intensity, distributed $\tau \sim Exp(0.1)$. The rain only affects the episode when the agent crosses the bridge: then, the agent suffers from an additive normally-distributed action noise $\mathcal{N}(0, \tau^2)$, in addition to a direct damage translated into a cost of $3 \cdot \tau$. For RoML, the CEM is implemented over the exponential family of distributions $Exp(\phi)$ with $\phi_0 = 0.1$ and $\beta = 0.05$. In this toy benchmark we use no regularization ($\nu = 0$).

In addition to the test returns throughout meta-training shown in Fig. 3, Fig. 8 displays the final test returns at the end of the meta-training, over 30 seeds and 3000 test tasks per seed.

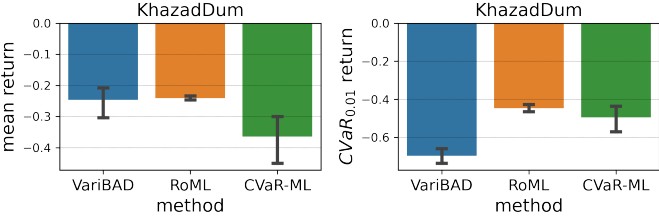

Figure 8: Khazad-Dum: Mean and CVaR returns over 30 seeds and 3000 test tasks.

## D.2  Continuous Control

In all the MuJoCo benchmarks introduced in Section 6.2, each task's meta-rollout consists of 2 episodes $\times$ 200 time-steps per episode. Below we describe the task distributions, as well as their parameterization for the CEM. In each benchmark, we used CEM quantile $\beta = 0.2$ and regularization of $\nu = 0.2$ samples per batch.

- **HalfCheetah-Vel**: The original task distribution is uniform $\tau \sim U([0, 7])$ in HalfCheetah (the task space $[0, 7]$ was extended in comparison to Zintgraf et al. [2019], to create a more significant tradeoff between tasks). We rewrite it as $\tau = 7\tilde{\tau}$, $\tilde{\tau} \sim Beta(2\phi, 2 - 2\phi)$ with $\phi_0 = 0.5$ (leading to $Beta(1, 1)$, which is indeed the uniform distribution). The CEM learns to modify $\phi$. Notice that this parameterization satisfies $\mathbb{E}_{D_\phi}[\tilde{\tau}] = \phi$.

- **Humanoid-Vel**: Same as HalfCheetah-Vel, with task space $[0, 2.5]$ instead of $[0, 7]$.

- **Ant-Goal**: The target location is random within a circle of radius 5. We represent the target location in polar coordinates, and write $r \sim Beta(2\phi_1, 2 - 2\phi_1)$ and $\theta \sim Beta(2\phi_2, 2 - 2\phi_2)$ (up to multiplicative factors 5 and $2\pi$). The original distribution parameter is $\phi_0 = (0.5, 0.5)$, and the CEM learns to modify it.

- **HalfCheetah-Mass, Humanoid-Mass, Ant-Mass**: The task $\tau \in [0.5, 2]$ corresponds to the multiplicative factor of the body mass (e.g., $\tau = 2$ is a doubled mass). The original task distribution is uniform over the log factor, i.e., $\log_2 \tau \sim U([-1, 1])$. Again, we re-parameterize the uniform distribution as $Beta$, and learn to modify its parameter.

- **HalfCheetah-Body, Humanoid-Body, Ant-Body**: The 3 components of the task correspond to multiplicative factors of different physical properties, and each of them is distributed independently and uniformly in log, i.e., $\forall 1 \leq j \leq 3 : \log_2 \tau_j \sim U([-1, 1])$. We re-parameterize this as 3 independent $Beta$ distributions with parameters $\phi = (\phi_1, \phi_2, \phi_3)$.

- **HalfCheetah 10D-task**: Again, the task components correspond to multiplicative factors of different properties of the model. This time, there are 10 different properties (i.e., the task space is 10-dimensional), but each of them varies in a smaller range: $\log_2 \tau_j \sim$

Table 2: Mean return over 1000 test tasks, for different models and MuJoCo environments. Standard deviation is presented over 30 seeds. CVaR returns are displayed in Table 1.

| | HalfCheetah | | | HalfCheetah 10D-task | | |
| | Vel | Mass | Body | (a) | (b) | (c) |
|---|---|---|---|---|---|---|
| CeSoR | $-1316 \pm 18$ | $1398 \pm 31$ | $1008 \pm 34$ | $1222 \pm 23$ | $1388 \pm 20$ | $1274 \pm 32$ |
| PAIRED | $-545 \pm 55$ | $662 \pm 30$ | $492 \pm 51$ | $551 \pm 53$ | $706 \pm 36$ | $561 \pm 65$ |
| CVaR-ML | $-574 \pm 22$ | $113 \pm 8$ | $193 \pm 6$ | $263 \pm 15$ | $250 \pm 11$ | $192 \pm 5$ |
| PEARL | $-534 \pm 15$ | $\mathbf{1726 \pm 13}$ | $\mathbf{1655 \pm 6}$ | $1843 \pm 9$ | $1866 \pm 13$ | $1425 \pm 6$ |
| VariBAD | $\mathbf{-82 \pm 2}$ | $1558 \pm 32$ | $1616 \pm 28$ | $\mathbf{1893 \pm 6}$ | $\mathbf{1984 \pm 67}$ | $\mathbf{1617 \pm 12}$ |
| RoML (VariBAD) | $-95 \pm 3$ | $1581 \pm 32$ | $1582 \pm 21$ | $1819 \pm 8$ | $\mathbf{1950 \pm 20}$ | $\mathbf{1616 \pm 13}$ |
| RoML (PEARL) | $-519 \pm 15$ | $1553 \pm 18$ | $1437 \pm 8$ | $1783 \pm 7$ | $1859 \pm 10$ | $1399 \pm 8$ |

| | Humanoid | | | Ant | | |
| | Vel | Mass | Body | Goal | Mass | Body |
|---|---|---|---|---|---|---|
| VariBAD | $\mathbf{880 \pm 4}$ | $\mathbf{1645 \pm 22}$ | $\mathbf{1678 \pm 17}$ | $-229 \pm 3$ | $1473 \pm 3$ | $\mathbf{1476 \pm 1}$ |
| RoML (VariBAD) | $\mathbf{883 \pm 4}$ | $1580 \pm 17$ | $1618 \pm 18$ | $\mathbf{-224 \pm 3}$ | $\mathbf{1475 \pm 2}$ | $1472 \pm 1$ |

$U([-0.5, 0.5])$. Each such property is a vector, and is multiplied by $\tau_j$ when executing the task $\tau$. The 10 properties are selected randomly, among all the variables of type *float ndarray* in *env.model*. We generate 3 such MRL environments – with 3 different random sets of 10 task-variables each. Some examples for properties are inertia, friction and mass.

In the experiments, we rely on the official implementations of VariBAD [Zintgraf et al., 2019] and PEARL [Rakelly et al., 2019], both published under the MIT license. CVaR-ML and RoML are implemented on top of these baseline, and their running times are indistinguishable from the baselines. All experiments were performed on machines with Intel Xeon 2.2 GHZ CPU and NVIDIA's V100 GPU. Each experiment (meta-training and testing) required 12-72 hours, depending on the environment and the baseline algorithm.

Table 2 and Fig. 9 present detailed results for our MuJoCo experiments, in addition to the results presented in Section 6.2.

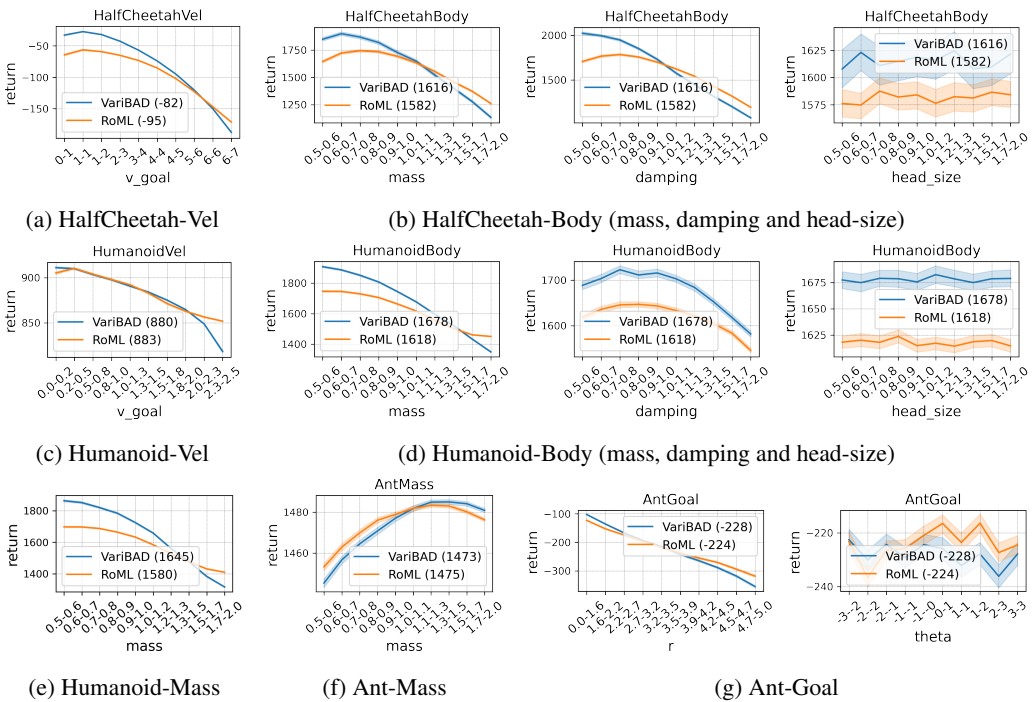

(a) HalfCheetah-Vel            (b) HalfCheetah-Body (mass, damping and head-size)

(c) Humanoid-Vel            (d) Humanoid-Body (mass, damping and head-size)

(e) Humanoid-Mass      (f) Ant-Mass      (g) Ant-Goal

Figure 9: Average return per range of tasks in the various MuJoCo environments (global average in parentheses). HalfCheetah-Mass is displayed in Fig. 4d.

## D.3   The Cross Entropy Method

We implemented RoML using the Dynamic Cross Entropy Method implementation of Greenberg [2022]. Below we concentrate results related to the CEM functionality in all the experiments:

- **Learned sample distribution**: One set of figures corresponds to the learned sample distribution, as measured via $\phi$ throughout the meta-training.
- **Sample returns**: A second set of figures corresponds to the returns over the sampled tasks (corresponding to $D_\phi$): ideally, we would like them to align with the $\alpha$-tail of the reference returns (corresponding to $D_{\phi_0}$). Thus, the figures present the mean sample return along with the mean and CVaR reference returns (the references are estimated from the sample returns using Importance Sampling weights). In most figures, we see that the sample returns (in green) shift significantly from the mean reference return (in blue) towards the CVaR reference return (orange), at least for part of the training. Note that in certain environments, the distinction between difficulty of tasks can only be made after the agent has already learned a basic meaningful policy, hence the sample returns do not immediately deviate from the mean reference.

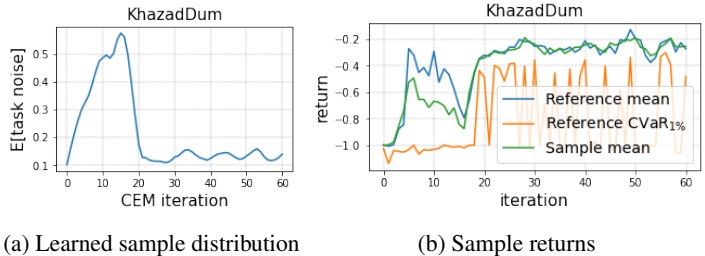

(a) Learned sample distribution      (b) Sample returns

Figure 10: The CEM in Khazad-Dum. Note that the effect of the CEM concentrates at the first half of the meta-training; once the meta-policy learns to focus on the long path, the agent becomes invariant to the sampled tasks, and the sampler gradually returns to the original task distribution $\phi \approx \phi_0$.

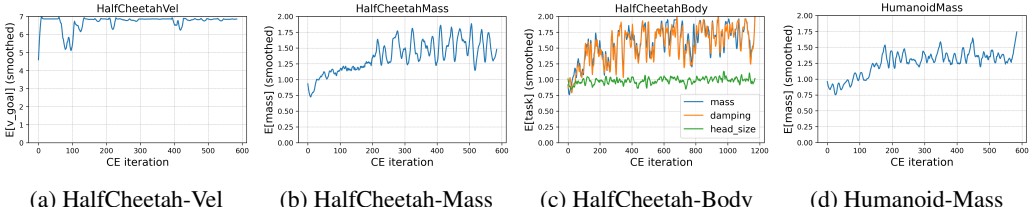

(a) HalfCheetah-Vel     (b) HalfCheetah-Mass     (c) HalfCheetah-Body     (d) Humanoid-Mass

Figure 11: Learned sample distribution in MuJoCo benchmarks. Notice that for HalfCheetah-Body, the CEM has to control 3 different task parameters simultaneously.

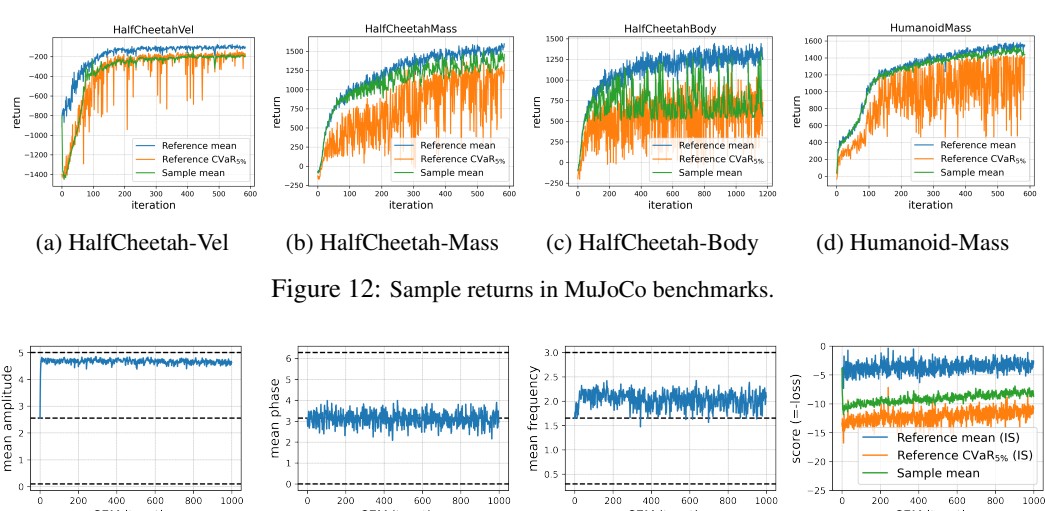

(a) HalfCheetah-Vel     (b) HalfCheetah-Mass     (c) HalfCheetah-Body     (d) Humanoid-Mass

Figure 12: Sample returns in MuJoCo benchmarks.

Figure 13: Sine Regression: The 3 left figures show the learned sample distribution, corresponding to average amplitudes, phases and frequencies. The CEM immediately learns that the amplitude has the strongest effect on test loss, whereas frequency has a moderate effect and phase has none. The right figure shows in green the mean sample scores (in supervised learning these are the negative losses instead of the returns).

## D.4 Ablation Test: RoML with a Naive Sampler

RoML learns to be robust to high-risk tasks by using the CEM to over-sample them throughout training. The dynamic CEM presented in Section 5 is designed to identify and sample high-risk tasks from a possibly-infinite task-space, where the task values change with the evolving policy throughout training.

In this section, we conduct an ablation test to demonstrate the importance of the CEM to this task. To that end, we implement a naive adversarial task sampler. The first $\mathcal{M} = 100$ tasks are sampled from the original distribution $D$, and the naive sampler memorizes the $\alpha\mathcal{M}$ lowest-return tasks, and samples randomly from them for the rest of the training.

As displayed in Table 3, switching to the naive sampler decreases the CVaR returns significantly.

Table 3: Ablation test: $CVaR_{0.05}$ return, compared to the naive sampler baseline.

| | HalfCheetah Body | HalfCheetah 10D-task (a) | (b) | (c) |
|---|---|---|---|---|
| VariBAD | $835 \pm 30$ | $1126 \pm 6$ | $1536 \pm 39$ | $988 \pm 13$ |
| Naive sampler | $839 \pm 20$ | $1056 \pm 34$ | $1340 \pm 57$ | $978 \pm 12$ |
| RoML (VariBAD) | $935 \pm 17$ | $1227 \pm 13$ | $1697 \pm 24$ | $999 \pm 20$ |

# E  Supervised Meta-Learning

Below we provide the complete details for the toy Sine Regression experiment of Section 6.3. The input in the problem is $x \in [0, 2\pi]$, the desired output is $y = A\sin(\omega x + b)$, and the task is defined by the parameters $\tau = (A, b, \omega)$, distributed uniformly over $\Omega = [0.1, 5] \times [0, 2\pi] \times [0.3, 3]$. Similarly to Finn et al. [2017], the model is fine-tuned for each task via a gradient-descent optimization step over 10 samples $\{(x_i, y_i)\}_{i=1}^{10}$, and is tested on another set of 10 samples. The goal is to find model weights that adapt quickly to new task data.

CVaR-ML and RoML are implemented with robustness level of $\alpha = 0.05$, on top of MAML. For the CEM of RoML, we re-parameterize the uniform task distribution using Beta distributions (see CEM details below).

As shown in Fig. 13, RoML learns to focus on tasks (sine functions) with high amplitudes and slightly increased frequencies, without changing the phase distribution. Fig. 14 displays the test losses over 30 seeds, after meta-training for 10000 tasks. Similarly to the MRL experiments, again RoML achieves better CVaR losses than both CVaR-ML and the baseline.

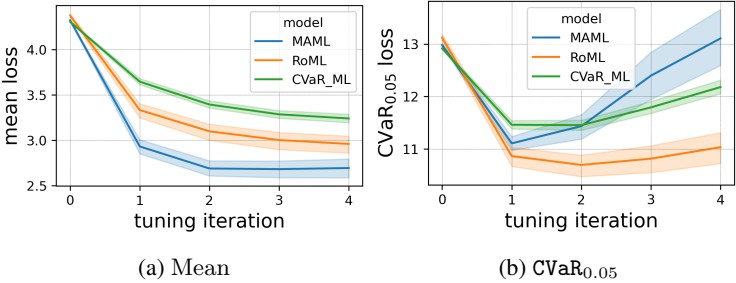

(a) Mean                                    (b) $CVaR_{0.05}$

Figure 14: Sine Regression: Mean and CVaR losses over 10000 test tasks, against the number of tuning gradient-steps at test time. The 95% confidence intervals are calculated over 30 seeds.

**CEM implementation details:**   In comparison to Finn et al. [2017], we added the sine frequency as a third parameter in the task space, since the original problem was to simplistic to pose a mean/CVaR tradeoff. The tasks are distributed uniformly, and we reparameterize them for the CEM using the Beta distribution, similarly to Appendix D.2: $\tau = \{\tau_j\}_{j=1}^3$, $\tau_j \sim Beta(2\phi_j, 2 - 2\phi_j)$. On top of this, we add a linear transformation from the Beta distribution domain $[0, 1]$ to the actual task range ($[0.1, 5]$ for amplitude, $[0, 2\pi]$ for phase and $[0.3, 3]$ for frequency). Note that the original uniform distribution is recovered by $\phi_0 = (0.5, 0.5, 0.5)$. The parameter $\phi_j \in [0, 1]$, which is controlled by the CEM, equals the expected task $\mathbb{E}_{\tau \sim D_\phi}[\tau_j]$. The other hyper-parameters of RoML are set to $\beta = 0.2$ (CEM quantile) and $\nu = 0$ (no regularization).

