# OpenReview forum: "Train Hard, Fight Easy: Robust Meta Reinforcement Learning"
_NeurIPS.cc/2023/Conference — NeurIPS 2023 poster_

### Official Review · Reviewer_FerX · 2023-07-02

**Soundness:** 4 excellent
**Presentation:** 4 excellent
**Contribution:** 4 excellent
**Rating:** 6
**Confidence:** 3

**Summary:**

This is an interesting paper that harnesses a risk-averse optimization from RL and applies the trick to Meta RL (called RoML). It has a theory section that proves that biased gradients in ordinary robust RL objectives are no longer biased when applied to meta RL, an interesting result. there is also a data inefficiency problem that arises in robust RL objectives because out of a batch of N trajectories, they can only make use of a small proportion (alpha) of them. The authors importance sample the set of tail tasks, recovering sample efficiency. Altogether, the authors discuss several key problems that are present in vanilla RL which disappear in meta RL.

The authors demonstrate experimentally on a variety of tasks the benefits of RoML over baselines.

**Strengths:**

This paper discovers the cool trick that whereas CVaR has problems associated with its biased estimate of gradients in ordinary RL, these problems are resolved in Meta RL, which is a cool finding backed by theory. The intuition is easy to understand. The experiments are solid.

**Weaknesses:**

I have no concerns. The authors demonstrate strong performance on a variety of tasks, demonstrating its versatility.

**Questions:**

N/A

---

> ### Author Rebuttal · Authors · 2023-08-08
>
> We thank the reviewer for reading our work, and are encouraged that the reviewer found it cool, intuitive and solid.

---

> > ### Comment · Reviewer_FerX · 2023-08-16
> >
> > Thanks for the reply, best of luck to the authors.

---

### Official Review · Reviewer_vMWn · 2023-07-03

**Soundness:** 2 fair
**Presentation:** 3 good
**Contribution:** 3 good
**Rating:** 6
**Confidence:** 4

**Summary:**

This paper focuses on the meta-reinforcement learning setting, where the agent learns a meta-policy that adapts to new tasks with few observations. However, traditional meta-RL methods optimize the average return over tasks but often suffer from poor results in tasks of high risk or difficulty. This may make the agent fail to generalize to risk tasks. In this paper, the authors introduce CVaR into the meta RL framework to solve this problem. They theoretically prove that an unbiased gradient estimator is available in the meta RL setting, which allows a general critic module. They then propose the Robust Meta RL algorithm (RoML) algorithm to generate a robust version of any given MRL algorithm, by identifying and oversampling harder tasks throughout training. Experiments on the gym locomotion tasks demonstrate the advantages of the proposed method.

**Strengths:**

1.	This paper is well-written and easy to follow. The figures are clear and illustrative for understanding the idea.
2.	The theoretical analysis shows evidence to support the advantages of the proposed method.
3.	The experiments also support the claim that the proposed method achieves better adaptation to unseen tasks.


**Weaknesses:**

1.	I am a little confused about the setting of robust meta-reinforcement learning. What does robustness mean in meta RL? Is there any disturbance to the dynamic system or observation? It seems that the setting mentioned in this paper is that some tasks are harder than others.
2.	The main contribution of this paper is extending the CVaR-PG framework to meta RL setting and eliminating the limitation of the biased gradient estimator. However, the intuition behind the difference between using CVaR in RL and meta-RL settings is not clear. If the task space is not discretized (as mentioned in the caption of Figure 1), the boundary of RL and meta RL will be unclear. So we can still have both easy and hard situations in the RL setting and assume there is an unknown distribution of the difficulty. I think the authors should give some hints in the introduction section about why using CVaR in meta RL is better than in RL. Does the advantage come from the discretized task indicator?
3.	The experimental evaluation is not thorough enough. First, the experiments do not use standard meta-RL benchmarks, such as Meta World. Second, the experimental setting is not consistent with the motivation “Standard MRL methods optimize the average return over tasks but often suffer from poor results in tasks of high risk or difficulty.” There are no particular high-risk or difficult tasks in the environments （a high-risk task is defined as the high mass in the mujoco environment）. The example in Figure 1 looks good but is not mentioned in the experiment part. Third, potential simple baselines are missing. For example, oversampling the hard task according to the return. This could be a general trick for all meta-learning methods, so I think it should be a fair baseline.


**Questions:**

Points 1 and 2 of the weaknesses are my questions.

**Limitations:**

The limitation of the proposed method is discussed in the conclusion section. The algorithm requires control over the selection of training tasks. This assumption is acceptable since it is possible to define and control different tasks in both simulation and real-world cases.

---

> ### Author Rebuttal · Authors · 2023-08-08
>
> We thank the reviewer for their detailed and helpful feedback. Please see our responses below, as well as the new experiments we added following the review.
>
> **What does robustness mean in meta RL?**
>
> We defined the robustness of an agent in meta RL as the CVaR return of the agent. High CvaR returns express robustness to the selection of tasks, as it results in a reasonable return even under adversarial tasks. This is analogous to robustness in RL, where we aim to perform well under adversarial deviation of the underlying model. In addition, interpreting the CVaR as robust optimization is supported by Chow et al. 2015: they showed that optimizing CVaR is equivalent to robust RL under a certain uncertainty set. To clarify our suggested notion of robustness, we will add this discussion to the intro, thanks!
>
> **Potential simple baselines are missing.**
>
> The submission contained two sample-based baselines, PAIRED [Dennis et al. 2020] and CeSoR [Greenberg et al. 2022]. Following this comment, **we now conducted experiments with an additional baseline, which will be added to the manuscript.** The naive sampler baseline memorizes the alpha% tasks of lowest returns, among the first batch of 100 sampled tasks, and then focuses the training on these lowest-return tasks. As presented below, this resulted in significantly lower CVaR returns:
>
> |                |     Cheetah-Body     |  Cheetah-10D-task (a) |  Cheetah-10D-task (b) | Cheetah-10D-task (c) |
> |:--------------:|:------------:|:-------------:|:-------------:|:------------:|
> | VariBAD        | $835 \pm 30$ |  $1126 \pm 6$ | $1536 \pm 39$ | $988 \pm 13$ |
> | Naive sampler  | $839 \pm 20$ | $1056 \pm 34$ | $1340 \pm 57$ | $978 \pm 12$ |
> | RoML           | $935 \pm 17$ | $1227 \pm 13$ | $1697 \pm 24$ | $999 \pm 20$ |
>
> We explain the advantage of RoML by addressing the following challenges:
> * Generalization to unseen tasks: in MuJoCo’s continuous task spaces, with infinite tasks, memorization of tasks is inherently limited. By contrast, the CEM modeling of RoML captures the structure of the task space, and learns which tasks are harder - even for tasks that have not been observed yet.
> * Adaptation to the ever-changing policy: the lowest-return tasks at the beginning of training do not necessarily remain so throughout the whole training. RoML’s CEM adapts continuously to the observed returns, and also keeps sampling some high-return tasks - in case that their returns will drop at some point.
>
> We can extend the experiments beyond the 4 environments presented above, if the reviewer considers it beneficial.
>
> **The experiments do not use standard meta-RL benchmarks, such as Meta World.**
>
> Our primary experiments follow and extend the standard MuJoCo MRL benchmarks from Zintgraf et al, 2019 (VariBAD). As discussed above, these MuJoCo benchmarks present infinite, continuous, but structured task spaces, which pose an additional challenge, as the low-return tasks cannot be learned by naive memorization. Thus, we found these environments more suitable for testing of RoML.
>
> **The experimental setting is not consistent with the motivation.**
>
> First, the toy Khazad-Dum benchmark does follow a disaster-prevention motivation, aiming not to fall off the bridge. Second, our general motivation is to minimize the risk of very low returns. RoML learns where this risk comes from. In some experiments, poor returns turned out to be caused by high body-mass, thus RoML learned to focus on such tasks, which we see as consistent with the motivation.
>
> **The difference between using CVaR in RL and meta-RL settings is not clear… Does the advantage come from the discretized task indicator?**
>
> In RL, when the policy changes, this affects both (a) the return of high-risk situations, and (b) the probability to get to these situations.
>
> In MRL, when the policy changes, this only affects (a) the task return, while (b) the task probabilities remain the same (since the task distribution does not depend on the policy).
>
> Since (b) is the cause for the gradient bias in RL (as illustrated in Figure 5 in Appendix A), the bias disappears in MRL (illustrated in Figure 6).
>
> This is the substantial difference between robustness in RL and in MRL, and it holds for both discrete and continuous task spaces.
> We will stress this root difference in the front manuscript, in addition to Appendix A.

---

> > ### Comment · Reviewer_vMWn · 2023-08-10
> > **Response to review**
> >
> > Thanks for addressing my concerns. I don't have further questions so I am happy to increase my score to weak accept.

---

### Official Review · Reviewer_CM4Y · 2023-07-04

**Soundness:** 3 good
**Presentation:** 3 good
**Contribution:** 3 good
**Rating:** 6
**Confidence:** 4

**Summary:**

The paper presents a novel approach to robust Meta-Reinforcement Learning (MRL), aiming to improve the consistency of returns across tasks, crucial for many RL applications. The authors introduce Conditional Value-at-Risk (CVaR) optimization to MRL, and propose a new method called Robust Meta RL algorithm (RoML). This approach overcomes issues of biased gradients, common in CVaR optimization in RL, and increases sample efficiency by identifying and over-sampling lower return tasks. Experimental results support the effectiveness of RoML, showing improved robustness over baseline algorithms in various domains.

**Strengths:**

**Strengths**:

1. The use of CVaR optimization in Meta RL is a novel idea that addresses the issue of robustness, a common challenge in reinforcement learning applications.

2. The theoretical proof that MRL is immune to biased gradients, a common issue in CVaR optimization in RL, is a significant strength.

3. Experimental results highlight the robustness and efficiency of the RoML method across various domains.

**Weaknesses:**

**Weaknesses**:

1. The RoML algorithm assumes that tasks can be selected during training. This might not always be feasible in real-world applications, potentially limiting its general applicability.

2. While the paper mentions that average return is improved in certain experiments, it does not provide a clear discussion on this, leaving questions on how RoML impacts average return across different applications.


**Questions:**

1. Could you elaborate on the conditions under which RoML improves the average return? Does this improvement come at the cost of performance in other aspects?

2. Could you provide more clarity on how task selection during training works with the RoML algorithm?

**Limitations:**

This paper includes a few sentences of the main limitations. I suggest authors discuss more on this.
I don't find any obvious societal impacts.

---

> ### Author Rebuttal · Authors · 2023-08-08
>
> We thank the reviewer for their helpful comments. Please see our responses below.
>
> **Could you elaborate on the conditions under which RoML improves the average return?**
>
> This result is empirical, and we specify it mostly to show that optimizing the mean and the CVaR are not necessarily contradicting. That is, sometimes low-return tasks can be improved at the cost of high-return tasks, but without hurting average performance. This leaves the question: why does the mean improve? We hypothesize that focusing on hard tasks forms a learning curriculum, which has been shown to help learning sometimes. We will discuss this hypothesis in the manuscript. Thanks!
>
>
> **Could you provide more clarity on how task selection during training works?**
>
> Task sampling relies on the Cross-Entropy Method (CEM). It is discussed in Section 5 and we also provide a simplified presentation including pseudo code in Appendix C1.
> The main idea is that we sample $N$ tasks and generate rollouts for them. Then, we take the $\beta\cdot N$ tasks with worst returns, and update the sample distribution accordingly (we define a class of task-distributions in advance, and fit a distribution from the class to these worst-returns tasks). Since $\beta>\alpha$, fitting $\beta\cdot N$ tasks is less prone to overfit. By repeatedly fitting the worst tasks of the *current* distribution, we eventually reach the extreme alpha-tail tasks of the *original* distribution, as desired.
>
> **I suggest authors discuss more on the limitations.**
>
> As suggested in the review, we will discuss the limitations of the methods in more detail. In particular, we will connect the discussion in Lines 242-246 with Lines 374-379, for a more coherent presentation of the limitations. Thanks!

---

> > ### Comment · Reviewer_CM4Y · 2023-08-15
> >
> > Thanks for the authors' clarification. My score remains the same as weak accept.

---

### Official Review · Reviewer_ib31 · 2023-07-07

**Soundness:** 3 good
**Presentation:** 3 good
**Contribution:** 3 good
**Rating:** 6
**Confidence:** 3

**Summary:**

 This paper aim to address the limitations in the standard meta reinforcement learning (MRL) methods, where they optimize the average return over tasks and often struggle with high-risk or difficult tasks, hence limiting system reliability. To this end, the authors define a robust MRL objective with a controlled robustness level to overcome these limitations. Meanwhile, this work proposed RoML with provable gradient bias reduction and improvement on the data efficiency. Overall, this paper is well-organized and presented. The proposed method is promising with experimental evidence. They show that RoML achieves robust returns, indicating improved performance and reliability in challenging scenarios. By combining the controlled robustness of the MRL objective and the oversampling approach of RoML, this paper contributes to advancing the field of RL in real-world applications with variable and unknown test tasks.

**Strengths:**

1. To address the impact of the biased gradient issue in standard RL, this work proposed to use CVaR in the objective and provides the theoretical guarantee (Theorem 1)
2. The proposed CVaR-ML has improved data efficiency by a factor up to $1/\alpha$
3. The conducted experiments on both risk-averse, risk-neutral baselines shows promising results of RoML.

**Weaknesses:**

1. The proposed algorithm RoML requires the proper set up of parameter $\alpha$ and $\beta$, it will be beneficial to discuss more on the choice of those parameters for different tasks? e.g., do you use grid-search for choosing those parameters?

**Questions:**

1. Does the rollout length matter to the performance? Do you choose different rollout length for different tasks?
2. Does over-sampling on the low-return tasks increase the computation complexity?
3. If the task distribution changes overtime, does the over-sampling strategy still be able to guarantee the robustness?

**Limitations:**

The author has discussed the limitations of this work.

---

> ### Author Rebuttal · Authors · 2023-08-08
>
> We thank the reviewer for their helpful comments. Please see our responses below.
>
> **RoML requires the proper set up of parameter alpha and beta.**
>
> The parameter alpha is part of the problem definition and it is not a hyper-parameter of RoML. It expresses the desired robustness level, which is analogous to the uncertainty-set size in standard robust RL.
>
> Beta is indeed a hyper-parameter and can be set using hyper-parameter tuning on a validation set. In practice, the value of $\beta=0.2$ (taken from the CEM of Greenberg et al. 2022) worked well for us. We will clarify this issue in the revised version.
>
> **Do you choose different rollout length for different tasks?**
>
> Both the number of episodes-per-task ($K$) and episode lengths are constant per benchmark in our work. For most benchmarks, we just set them according to the MuJoCo benchmarks of VariBAD (Zintgraf et al, 2019).
>
> **Does over-sampling on the low-return tasks increase the computation complexity?**
>
> The additional complexity is negligible. We oversample low-return tasks **instead** of other tasks, so  RoML does not require additional samples (this is the key advantage of RoML over CVaR-ML). The task selection itself relies on the CEM, whose additional running-time overhead is negligible (see Line 304).
>
> **If the task distribution changes overtime, does the over-sampling strategy still be able to guarantee the robustness?**
>
> To some extent. The CEM searches for low-return tasks in a given task distribution $D$. It yields the CEM’s sample-distribution $D_\phi$. You could change $D$ to some $D’$ in the middle of training, and the CEM would adapt to $D’$ in time (via the importance-sampling weights) – as long as $D$, $D’$ and $D_\phi$ all have the same support. Notice that in Appendix D, we indeed defined all the CEM distribution-families to have constant supports. Finally, if $D$ changes repeatedly, the CEM may still catch up if the change is slower than the CEM’s convergence rate. This is an interesting problem for future work, and we will add this discussion to the last section. Thanks!

---

> > ### Comment · Reviewer_ib31 · 2023-08-10
> > **Thank you for the rebuttal**
> >
> > Thank you for the effort on the rebuttal.
> >
> > - Q1 on rollout length: It is feasible to use the same setting as previous work to set up the rollout length while the author need to give explicit setup at least in the experiments. Meanwhile, it is well known that the rollout plays a critical part on the learning performance and data efficiency. The authors need to specify how does the choice of rollout length have impact on the results?

---

> > > ### Author Response · Authors · 2023-08-17
> > >
> > > Thanks for clarifying the question. First, we will add a report of the rollout lengths to the manuscript - both number of episodes per task, and number of steps per episode.
> > >
> > > Second, as requested, we tested the sensitivity to the rollout length by modifying the number of episodes K per task. As shown below, in most experiments RoML obtains significantly superior CVaR.
> > >
> > > |               |         |  Cheetah-Vel  |  Cheetah-Mass | Cheetah-Body | Cheetah-10D (a) |
> > > |---------------|---------|:-------------:|:-------------:|:------------:|-----------------|
> > > | K=1           | VariBAD | $-243 \pm 23$ | $1089 \pm 21$ | $752 \pm 27$ | $1013 \pm 17$   |
> > > | K=1           | RoML    | $-258 \pm 42$ | $1253 \pm 22$ | $780 \pm 26$ | $1049 \pm 20$   |
> > > | K=2 (default) | VariBAD | $-202 \pm 6$  | $1072 \pm 16$ | $835 \pm 30$ | $1126 \pm 6$    |
> > > | K=2 (default) | RoML    | $-184 \pm 4$  | $1259 \pm 19$ | $935 \pm 17$ | $1227 \pm 13$   |
> > > | K=4           | VariBAD | $-225 \pm 23$ | $1108 \pm 20$ | $852 \pm 14$ | $1071 \pm 16$   |
> > > | K=4           | RoML    | $-235 \pm 37$ | $1275 \pm 22$ | $932 \pm 16$ | $1233 \pm 26$   |
> > >
> > > Third, regarding the choice of lengths: while in principle this can be considered a hyperparameter and be tuned, our approach for the experiments was different. We wanted to stress the effectiveness of RoML as a meta-algorithm that can take an existing algorithm and improve its robustness (Lines 83, 300) - without the user having to re-tune all its hyperparameters. To that end, we ran the baselines with standard hyperparameters from previous works, and used the *same* hyperparameters for RoML. This is an important aspect of our experimental approach and we will stress it further in the manuscript.

---

> > > > ### Comment · Reviewer_ib31 · 2023-08-17
> > > > **Thanks for your response**
> > > >
> > > > I thank the authors for the experimental results and discussions on my questions. I will keep my original score.

---

### Author Rebuttal · Authors · 2023-08-08

We are excited that all the reviewers recommended accepting the paper, and thank them for their helpful comments.

The reviewers found that our work **“contributes to advancing RL in real-world application”** (ib31) and is **“crucial for many RL applications”** (CM4Y). They consider our **theoretical guarantees** in providing unbiased gradients a **”significant strength”** and a **”cool finding”** (CM4Y, FerX). The **improved sample efficiency** is also highlighted (ib31, CM4Y, FerX). The reviewers found that our experiments are **“solid”** (FerX) and **“highlight the robustness and efficiency”** (ib31, CM4Y), and concluded that our method **”achieves better adaptation”** (vMWn) – on **“a variety of tasks”** (CM4Y, FerX) including **”challenging scenarios”** (ib31). Finally, the reviewers have found the paper **“well-written”** (ib31, vMWn).

Our responses to the reviewers’ comments are below.

---

### Decision · Program_Chairs · 2023-09-21

**Decision:**

Accept (poster)

**Comment:**

**Summary:** This paper presents a novel formulation of robust meta-RL, where the standard objective of expectation (of the value) over the tasks is replaced by a CVaR objective. This, however, immediately poses computational challenges.  In particular, if one has to use policy optimization methods, then a well-defined policy gradient that can be estimated from trajectory samples is needed. The first main contribution of this paper is such a meta policy gradient theorem. The paper then proposes a variance reduction scheme, and a practical algorithm called Robust Meta RL (RoML). The paper presents experimental results that illustrates the performance of the proposed algorithm.

**Comments:** The review scores are 6, 6, 6, 6, and average is 6.  The paper is technically solid, and gives meta policy gradient theorem, a variance reduction approach, and a practical algorithm called Robust Meta RL (RoML). The paper presents experimental results that illustrates the performance of the proposed algorithm by comparing it against a non-robust meta RL algorithms, and (non-meta) risk averse algorithms. Most of the reviewer comments were regarding the experiments and baseline, and the authors have given detailed explanation during rebuttal, including the addition of another baseline. During the discussion period, the reviewers did not raise any additional comments.

Please revise the paper by including the all the changes you promised during the rebuttal, such as additional simulations, experiments with new baselines, more clarification about certain aspect, and a discussion about the limitations.

Thank you!

AC